# Sustained activation of the Aryl hydrocarbon Receptor transcription factor promotes resistance to BRAF-inhibitors in melanoma

Sébastien Corre[1], Nina Tardif[1], Nicolas Mouchet[1], Héloïse M. Leclair[1], Lise Boussemart[1,2], Arthur Gautron[1], Laura Bachelot[1], Anthony Perrot[1], Anatoly Soshilov[3], Aljosja Rogiers[4,5], Florian Rambow[4,5], Erwan Dumontet[6], Karin Tarte [6], Alban Bessede[7], Gilles J. Guillemin [8], Jean-Christophe Marine[4,5], Michael S. Denison[3], David Gilot [1] & Marie-Dominique Galibert [1,9]

BRAF inhibitors target the BRAF-V600E/K mutated kinase, the driver mutation found in 50% of cutaneous melanoma. They give unprecedented anti-tumor responses but acquisition of resistance ultimately limits their clinical benefit. The master regulators driving the expression of resistance-genes remain poorly understood. Here, we demonstrate that the Aryl hydrocarbon Receptor (AhR) transcription factor is constitutively activated in a subset of melanoma cells, promoting the dedifferentiation of melanoma cells and the expression of BRAFi-resistance genes. Typically, under BRAFi pressure, death of BRAFi-sensitive cells leads to an enrichment of a small subpopulation of AhR-activated and BRAFi-persister cells, responsible for relapse. Also, differentiated and BRAFi-sensitive cells can be redirected towards an AhR-dependent resistant program using AhR agonists. We thus identify Resveratrol, a clinically compatible AhR-antagonist that abrogates deleterious AhR sustained-activation. Combined with BRAFi, Resveratrol reduces the number of BRAFi-resistant cells and delays tumor growth. We thus propose AhR-impairment as a strategy to overcome melanoma resistance.

[1] IGDR (Institut de Génétique et Développement de Rennes)—UMR6290, CNRS, Univ Rennes, F-35000 Rennes, France. [2] Department of Dermatology, Hospital University of Rennes (CHU Rennes), F-35000 Rennes, France. [3] Department of Environmental Toxicology, University of California, Meyer Hall, Davis, CA 95616, USA. [4] Laboratory for Molecular Cancer Biology, VIB Center for Cancer Biology, VIB, Leuven 3000, Belgium. [5] Laboratory for Molecular Cancer Biology, Department of Oncology, KU Leuven, Leuven 3000, Belgium. [6] MICMAC (MIcroenvironment, Cell differentiation, iMmunology And Cancer) —UMR_S 1236, Inserm, Univ Rennes, F-35000 Rennes, France. [7] ImmuSmol, Pessac F-33600, France. [8] Neuroinflammation Group, MND and Neurodegenerative Diseases Research Center, Macquarie University, Sydney, NSW 2109, Australia. [9] Department of Molecular Genetics and Genomics, Hospital University of Rennes (CHU Rennes), F-35000 Rennes, France. These authors contributed equally: Sébastien Corre, Nina Tardif. These authors jointly supervised this work: Sébastien Corre, David Gilot, Marie-Dominique Galibert. Correspondence and requests for materials should be addressed to S.C. (email: sebastien.corre@univ-rennes1.fr) or to D.G. (email: david.gilot@univ-rennes1.fr) or to M.-D.G. (email: mgaliber@univ-rennes1.fr)

**B**RAF inhibitors (BRAFi) target selectively the BRAF V600E/ K genetic alteration found in several cancers. Cutaneous melanoma, the most aggressive form of skin cancer, harbor the highest incidence of this mutation (50%)[1,2]. Development of BRAFi in melanoma has thus served as a model for their implementation, revolutionizing personalized medicine. They give an impressive but transient response since resistance ultimately limits their clinical benefit[3–6]. The efficacy of BRAFi is indeed limited by intrinsic/primary mechanisms and/or acquired/ secondary resistances[7]. Besides these well describe genomic alterations that mainly promote the reactivation of the MAPK and/or the PI3K-signaling, activation of BRAFi-resistant gene (AXL, EGFR…) constitutes an additional hallmark of resistance[8,9]. Importantly, it has recently been shown that acquisition of these BRAFi resistance programs arise in a subset of melanoma cells and is associated with a dedifferentiated status of the melanoma cells[10,11]. Together, this increases the complexity and fosters the identification of the master regulators driving the expression of these resistance-genes that remain still unknown[12–17]. Here, we mainly focus on the potential role of AhR transcription factor in resistance mechanisms occurring during melanoma treatment by BRAFi.

The Aryl hydrocarbon Receptor (AhR) is a ligand-dependent transcription factor of the basic-helix-loop-helix (bHLH) Per-Arnt-Sim (PAS) family. Exogenous and endogenous binding-ligands, such as TCDD (2,3,7,8-tetrachlorodibenzo-p-dioxin) and FICZ (5,11-dihydroindolo[3,2-b]carbazole-6-carboxaldehyde), respectively[18], promote AhR translocation into the nucleus. In the nucleus, AhR dimerizes with the AhR nuclear translocator (ARNT), forming a DNA binding complex that binds and activates the transcription of target genes that harbor xenobiotic responsive elements (XREs). AhR agonists thereby induce the expression of, among others, the drug-metabolizing cytochrome P-450 (CYP) enzymes *CYP1A1, CYP1B1*, and *TIPARP*[19]. *CYP1A1* is commonly considered a prototypical AhR target[20]. Increasing evidence indicates that besides its roles in detoxification, AhR is involved in many physiological processes[21,22], diseases, and cancers[23].

In this study, we established an important role of AhR transcription factor in controlling sensitivity or resistance to BRAFi in melanoma. In tumor cells, BRAFi constitute new AhR ligands promoting melanoma sensitivity while a small subpopulation of cells has a high canonical AhR activity that is responsible for resistance acquiring and relapse. We also demonstrated that AhR constitutes a therapeutic target to delay relapse during the treatment of melanoma by BRAFi and thus merits to be tested in human. Together, this study contributes to the understanding of the molecular mechanisms driving BRAFi resistance and relapse, and proposes a therapeutic combination to overcome these deleterious effects.

## Results

### BRAFi as new AhR ligands controlling its transcriptional activity.
We observed that the BRAFi Vemurafenib (Vem) binds directly to AhR and stimulates its nuclear translocation (Fig. 1a, b). However, surprisingly, in contrast to TCDD (Fig. 1d), Vem failed to stimulate the canonical AhR/ARNT-XRE pathway after dimerization with ARNT (Fig. 1c). Consequently, Vem failed to induce endogenous *CYP1A1* expression (Fig. 1e) and CYP1A enzymatic activity (EROD) as observed with TCDD (Fig. 1f). These results indicated that Vem binds to AhR differently than canonical AhR ligands. Consistently, docking experiments have demonstrated that Vem and the canonical AhR ligand/agonist TCDD interact with AhR at different positions (Fig. 1g). The Vem and canonical AhR

ligand binding positions will be hereafter referred as the β- and α-pockets, respectively.

Importantly, non-canonical binding to AhR was observed with other BRAFi, including Dabrafenib (Dab) (Supplementary Figure 1), indicating that this is not a specific property of Vem only but rather of this chemically related family of molecules[24]. To investigate the molecular consequences of BRAFi-recruitment on AhR, we established the transcriptomes of BRAF-V600E melanoma cells (501Mel) exposed to Vem and to TCDD. Interestingly, specific and mutually exclusive signatures were identified (Fig. 1h). Consistent with the ability of Vem to increase pigmentation in cultured melanoma cells (Fig. 1i) and, in some patients' nevi (Supplementary Figure 2a), the Vem-induced signature was enriched in MITF-targets and in pigmentation genes (Fig. 1h)[25,26]. This signature, which significantly overlaps with the classical melanoma proliferative signature[14,15], was only observed with β-pocket ligands (Vem, Dab) and not with α-pocket ligands such as TCDD and Benzo(a)pyrene.

The pigmentation gene *Oculocutaneous Albinism Type II* (*OCA2*)[27] was particularly induced in Vem-exposed cells, and its level of expression was selected as a readout for AhR β-activation (Supplementary Figure 2a)[28]. Critically, genetic and chemical inhibition of AhR abrogated Vem-induced *OCA2* expression and subsequent induction of pigmentation (Supplementary Figure 2c-e). As anticipated from data reported in Fig. 1, ARNT was not required for Vem-induced *OCA2* expression, while ARNT depletion impaired TCDD-induced *CYP1A1* expression (Supplementary Figure 2h). Since off-target effects of Vem have been so far attributed to the paradoxical activation of the MAP Kinase pathway[24], we investigated the role of the MAPK pathway in the Vem-induced β-signature. We showed that the Vem-induced β-signature is independent of the phosphorylated status of ERK (Supplementary Figure 3) and is maintained in the presence of MEKi (Supplementary Figure 4a-c). Thus, the Vem-induced β-signature cannot be ascribed to BRAFi-induced paradoxical MAPK activation.

Due to the relative proximity between the α- and β-pockets, we hypothesized that ligands targeting the α-pocket could prevent the effect of molecules binding the β-pockets such as Vem. To illustrate this point, cells were exposed to an AhR-agonist or -antagonist (TCDD and CH-223191, respectively) that targeted the α-pocket, prior to Vem treatment. As envisaged, the occupancy of the α-pocket by either an AhR agonist or antagonist prevented Vem-associated effects (Supplementary Figure 4d-g). Reciprocally, Vem prevented the binding of TCDD in the α-pocket, demonstrating that Vem acts as an AhR antagonist (Supplementary Figure 4f). We further illustrated this latter capability by quantifying the AhR–ARNT/XRE complex formation that was reduced in the presence of Vem (Supplementary Figure 4g).

### AhR directs dedifferentiation and resistance to BRAFi.
The sensitivity of melanoma cells to BRAFi is associated with cellular differentiation state (i.e., MITF^high or pigmentation signature)[11,25,26]. Considering AhR-activation modulates the pigmentation program in melanoma cells, we investigated whether the AhR gene expression program is connected to BRAFi resistance[3–6]. We measured the transcriptional activity of both AhR canonical and non-canonical in melanoma cell lines from the "Cancer Cell Line Encyclopedia" (CCLE)[29], using representative 14 and 19 genes of the β- and α-signatures, respectively, and 16 genes from the BRAFi-resistance signature[13–16,25,30]. Whereas the β-signature was highly represented in the proliferative and BRAFi-sensitive cell lines, the α-signature was most prominent in BRAFi-resistant lines (Fig. 2a) and co-occurred with the resistance signature

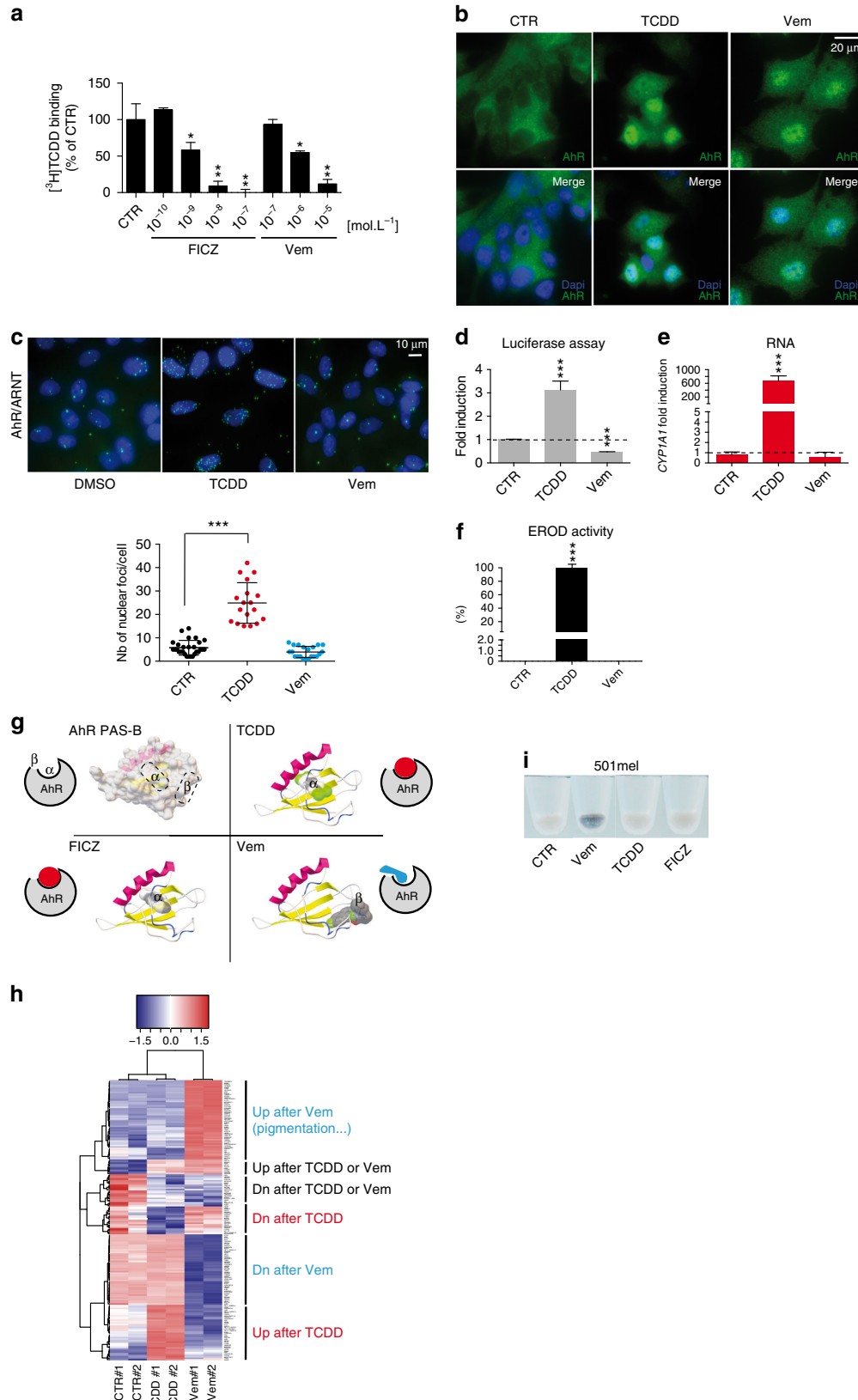

(Fig. 2b)[15,29]. In accordance with the recent classification of melanoma cells based on their differentiation states (melanocytic, M; transitory, T; neural crest-like, N; and undifferentiated cells, U), we confirmed that BRAFi-resistant cells are mostly dedifferentiated[10]. Moreover, the resistance-signature significantly overlapped with the invasive gene expression signature[15] and dedifferentiated cell state (Fig. 2c)[10,15,16,30,31].

To challenge these observations, we first re-analyzed the recently published Graeber's melanoma data sets[10] using their online webtool characterizing melanoma subtypes (M, T, N, or U)

**Fig. 1** BRAF-V600E inhibitor Vemurafenib binds to AhR and antagonizes the canonical AhR signaling pathway. **a** Competitive binding of FICZ or Vemurafenib (Vem) to AhR. Hepatic cytosol containing AhR was incubated with [$^3$H]TCDD in the presence of DMSO (1%) or increasing concentrations of FICZ ($10^{-10}$-$10^{-7}$ mol/L$^{-1}$) and Vem (PLX4032, $10^{-7}$-$10^{-5}$ mol/L$^{-1}$). **b** AhR nuclear translocation in response to Vem (1 µM) or TCDD (10 nM) in MCF-7 cells. AhR in green (IHC) and nucleus staining in blue. **c** AhR does not dimerize with ARNT in response to Vem (1 µM), in contrast to TCDD (10 nM), in MCF-7 cells. AhR–ARNT interaction was quantified by Proximity Ligation Assay. Hoechst-stained nucleus in blue ($n = 4$). **d–f** Vem does not activate the canonical transcriptional AhR response, in contrast to TCDD. **d** Evaluation of AhR transcriptional activity related to AhR/ARNT binding sites (XRE) using p3XRE-luciferase constructs. MCF-7 cells were exposed to 10 nM TCDD or 1 µM Vem or vehicle (DMSO) for 6 h. **e** Vem does not induce *CYP1A1* mRNA, in contrast to TCDD. MCF-7 cells were incubated in the absence or presence of 10 nM TCDD or 1 µM Vem for 15 h. **f** Vem does not induce EROD activity in contrast to TCDD. MCF-7 cells were either untreated or treated with 10 nM TCDD or 1 µM Vem for 6 h. **g** Proposed binding mode of TCDD, FICZ, and Vemurafenib (Vem) into the AhR PAS-B ligand binding domain homology model. Free binding energy is reported in Supplementary Table 1. The two predictive ligand binding pockets are indicated by (α) or (β). **h** Gene expression profile of the 501Mel cells exposed to vehicle, Vem (1 µM) or TCDD (10 nM) ($n = 2$) for 48 h. Heatmap focused on differentially expressed genes in function of treatment (fold change). **i** Vem induces pigmentation in vitro. Picture of 501Mel cell pellets treated with Vem (1 µM) or canonical AhR agonists TCDD (10 nM) or FICZ (1 µM) for 48 h. Data correspond to the mean ± s. d. of three independent experiments. Statistical analysis was performed using an unpaired *t*-test (PRISM6.0®) *$p < 0.05$; **$p < 0.01$; ***$p < 0.001$

(Fig. 2d) and showed that the β-signature decorates differentiated melanoma (M or T) cells (Fig. 2e) while the α- (Fig. 2f) and resistance signatures (Fig. 2g) are associated with dedifferentiated cell states (N and U). Second, using an additional set of melanoma cells lines that include three pairs of BRAFi sensitive/resistant cells (S and R), we demonstrated that acquisition of BRAFi resistance correlates with an increase in the α- and resistance signatures as well as a concomitant decrease in the β-signature (Fig. 2h). Together these results, suggest that AhR endorses this β- to α-signature shift. Interestingly, BRAFi/MEKi double blockade led to similar β- to α-reprogramming (Supplementary Figure 5)[32].

**AhR signatures classify patients' tumors**. Having established that AhR-signatures discriminate cell differentiation states and sensitivity/resistance to BRAFi, we explored these AhR signatures in melanoma samples. We first examined melanoma samples from the TCGA cohort[29]. Interestingly, we found that melanoma samples, naive of BRAFi treatment, segregated according to the α- and β- signatures (Fig. 3a). Almost 9% of these clinical biopsies showed a marked expression of the β-signature and 8% the α- and resistance-signatures in bulk analyses, thereby supporting the notion that the nature of the patient's response to BRAFi and time before relapse may be in part predefined (Fig. 3b). Again, we found that in these melanoma samples, the β-signature decorated differentiated melanoma samples (M and T) and the α- and resistance-signature undifferentiated ones (N and U) (Fig. 3d–f).

Second, we investigated the β- to α-signature shift in patients exposed to single or double drug-blockade (BRAFi or BRAFi + MEKi) by classifying their melanoma biopsies during the medication course (before, during, and at relapse) according to differentiated states (U, N, T, M) (Fig. 3g) and AhR-associated signatures (β-, α-, and resistance) (Fig. 3h). As anticipated from former TCGA results and recent studies, melanoma samples naive of any treatment (CTR) are characterized by an elevated intra- and inter-tumoral heterogeneity in terms of differentiation states and AhR signatures (Fig. 3g, h). For example, tumor no. 17 illustrates a melanocytic subtype (about 26% in the TCGA cohort) whereas tumor no. 9 a transitory subtype (about 63% in the TCGA cohort). Now, when exposed to drugs (BRAFi alone or BRAFi/MEKi) and according to the initial differentiation state, we observe either a strong β-signature (P1: Pt −17 −19 −2) followed by a switch to the α-signature (P2) or an immediate α-switch (P1: Pt −15 −9 122 −8). This immediate α-switch occurs for marked β-signature tumors (Fig. 3h). Again, the appearance of the α-signature co-occurred with the resistance signature and dedifferentiation process (Fig. 3i), as observed in melanoma cell lines

(Fig. 2e–h), supporting these AhR-dependent activation programs.

**AhR controls BRAFi resistance in melanoma persister cells**. It is noteworthy that the T, N, and U emerging subtypes arise within tumors in the presence of BRAFi (dotted line in Fig. 3i), suggesting that either the protective capacity of BRAFi in maintaining cells in a melanocytic state is overcome or that small and almost undetectable, but initially present, T, N, and U subpopulations emerge while melanocytic and BRAFi-sensitive populations diminish[10,11]. To characterize these subpopulations further, we investigated publicly available single-cell analyses that were more informative than bulk analyses[33] (Fig. 4a). As anticipated, only a small subpopulation of melanoma cells expressed a combination of BRAFi-resistance markers (EGFR, ZEB1…) and an α-signature. These cells (10%) are representative of N or U states (Fig. 4b, c). Now examining, cell-sorted EGFR-positive cells that are able to form colonies in the presence of a BRAFi[16] (Fig. 4d), we found that this minor subpopulation corresponds to dedifferentiated states of melanoma (N or U) expressing α- and resistant signatures (Fig. 4c–e). In contrast, the EGFR-negative cells exhibited a β-signature (M or T).

Together, these data suggest that BRAFi-resistant cells represent an innate/intrinsic small subpopulation of α-cells, generally undetectable through bulk analyses. These resistant cells become predominant in drug-resistant lesions/cultures upon drug-exposure probably due to cell death of sensitive β-cells (Fig. 5a)[16].

Our results also indicate that AhR signaling is maintained in a constitutively activated α-state in resistant cells. Since we did not find any recurrent mutation that may support constitutive activation of AhR in the TCGA melanoma cohort (cBioPortal: http://www.cbioportal.org/index.do), we hypothesized that endogenous α-ligands produced in drug-resistant cells may result in sustained activation of the AhR/ARNT pathway (Fig. 5b) and thereby promote the α-signature. Consistent with this possibility, we observed that the fate of β-cells could be redirected into α-cells upon exposure to α-ligands (TCDD), increasing the BRAFi-resistance gene signature (Fig. 5c), and that AhR drives the expression of BRAFi-resistance genes (*AXL* and *NRP1* among others) via the binding of promoter containing XRE motifs (Supplementary Figure 6). Finally, loss-of-function experiments confirmed that α- and resistance signatures were both AhR- and ARNT-dependent (Fig. 5e, f).

**AhR as a therapeutic target to delay resistance to BRAFi**. To exploit the potential therapeutic implications of these findings, we

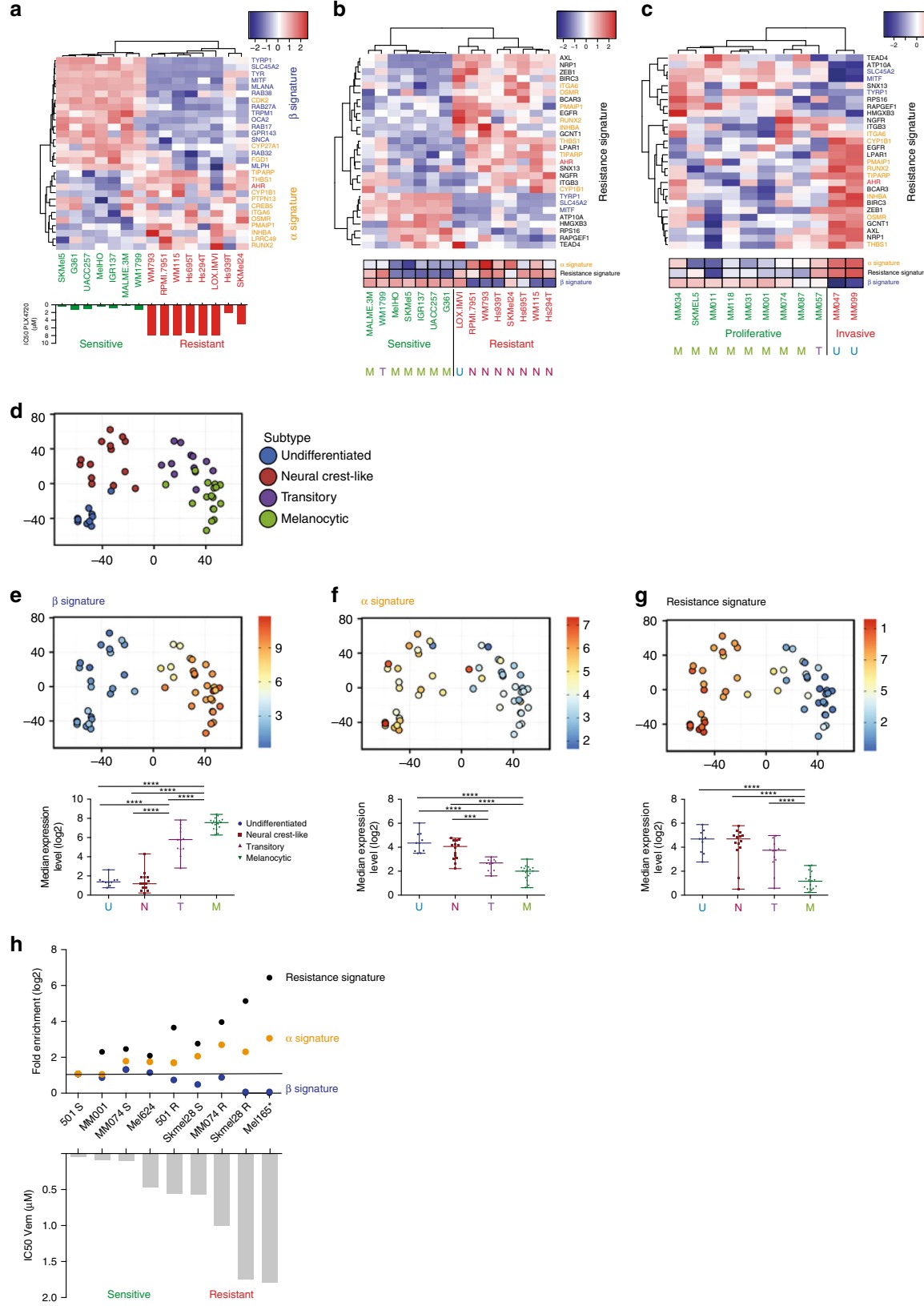

screened for molecules that impair deleterious α-activation of AhR (Fig. 6a). To this end, we performed in silico modeling combined with in vitro biochemical assays using a wide range of well-characterized AhR ligands, including exogenous and endogenous agonists (i.e., TCDD & B(a)P and FICZ & L-kynurenine,

respectively) and antagonists (i.e., CH-223191 or Resveratrol, RSV) (Fig. 6 and Supplementary Figure 7a). These analyses identified the well-tolerated molecule RSV, which selectively binds the α-pocket of AhR[34] (Fig. 6b). Critically, RSV-TCDD co-treatment failed to induce *CYP1A1* (Fig. 6c) and several other

**Fig. 2** AhR signature correlates with dedifferentiation states of melanoma cell lines and resistance to BRAFi. **a** Expression heatmap for β- and α-signature genes in different Vem-sensitive or -resistant melanoma cell lines from the Cancer Cell Line Encyclopedia RNAseq dataset (GEO, GSE36134[29]). IC50 values for PLX4720 were obtained from Supplementary Table 7 of ref. [29]. Genes and clusters with similar expression profiles across the cohort are placed close to each other in the grid. **b** Expression heatmap for BRAFi resistance genes in different Vem-sensitive or -resistant melanoma cell lines from Cancer Cell Line Encyclopedia RNAseq dataset (GEO, GSE36134[29]) (top) and average signatures for the α- (established by the median of expression of AhR target genes: *INHBA, THBS1, RUNX2, REEP2, PMAIP1, OSMR, LRRC49,* and *CYP1B1*), for the β- (established by the median of expression of pigmentation genes: *GPR143, TYR, SLC45A2, RAB38, SNCA, MLPH, MLANA,* and *MITF*), and for the resistance genes (*AXL, GCNT1, NRP1, ZEB1, ITGA1,* and *LPAR1*) (mid). Differentiation status for melanoma cell lines consistent with the four-stage differentiation model (melanocytic: M, transitory: T, neural crest-like: N, and undifferentiated: U)[10] has been established for the different melanoma cell lines considering average signature from subtype genes described in supplemented files from Tsoi et al.[10] (bottom). **c** Expression heatmap for β-, α-, and resistance genes signatures in the melanoma cultures dataset from the GEO dataset (GSE60664[15]) depending on their proliferative or invasive states. Average signatures and differentiation status for melanoma cell lines have been indicated at the bottom. **d** PCA of melanoma cell line datasets obtained by the cluster prediction assignment (from melanoma dedifferentiation signature resource from Graeber's lab: http://systems.crump.ucla.edu/dediff/)[10]. **e**–**g** *MLANA, PMAIP1,* and *AXL* expression PCA color maps illustrating, respectively, β- (**e**), α- (**f**), and resistance signatures (**g**) from different subtypes of melanoma cell lines dataset from Graeber's lab (top). Boxplots of selected β-genes (as described above) in different subtypes of melanoma cell lines (U, undifferentiated; N, neural crest-like; T, transitory; M, melanocytic) (bottom). Number in each group: U = 10, N = 14, T = 12, M = 17. Whiskers reflect median of expression with range. One-way ANOVA and Tukey's test: ***$p < 0.001$, ****$p < 0.0001$. **h** Fold expression level (log2) for average β-, α-, and resistance genes signatures in different melanoma cell lines from the lab compared to the 501Mel cell line (top). Vem sensitivity has been established by cell density measurement and calculation of the IC50 (bottom) using GraphPad (PRISM6.0®) 4 days after every 2 days of treatment with an increasing concentration of Vem. Asterisk: NRAS mutant

genes (Fig. 6d, Supplementary Figure 7b) of the α-signature, illustrating the effectiveness of RSV to counteract α-ligands[35]. RSV also inhibited BRAFi-induced pigmentation and induction of the β-signature, confirming the ability of α-ligands to compete with β-ligands for AhR binding (Fig. 6 and Supplementary Figures 3 and 8). Importantly, the sensitivity to BRAFi was fully maintained in RSV-exposed cells, indicating that this molecule does not alter BRAFi efficacy (Fig. 6e–h and Supplementary Figure 8). Together, these data indicate that an RSV-Vem combination treatment may result in a significant benefit over Vem alone. To test this possibility, we exposed pairs of BRAFi-sensitive (S) and resistant (R) melanoma cells (501Mel and SKMel28) to both Vem and RSV. RSV increased the sensitivity of sensitive and resistant cells to Vem (IC50, Fig. 6i, j) and decreased the number of BRAFi-persister cells (Fig. 6i–k). We evaluated the potential of this BRAFi/RSV combination in a patient-derived xenograft (PDX) model of melanoma[11,36]. Mice were exposed to BRAFi alone or in combination with RSV once tumors reached 200 mm³. In this model, BRAFi alone stabilized the tumor growth for about 10 days before relapse (Supplementary Figure 8d). BRAFi/ RSV treatment reduced, moderately but significantly, tumor growth compared to BRAFi alone (14 days) (Fig. 6l). In agreement with these results, tumors exposed to BRAFi/RSV combination reached endpoints significantly later than BRAFi alone (24 and 16 days, respectively) (Fig. 6m). Together, these results support the use of AhR inhibitors to increase BRAFi efficacy over-time.

## Discussion

In conclusion, we uncovered a central role for AhR in BRAFi resistance and relapse. Unexpectedly, we revealed that in drug-sensitive melanoma cells, BRAFi, in addition to inactivated oncogenic BRAF activity, functions as a non-canonical ligand of AhR. Consequently, BRAFi promotes an AhR-transcriptional pathway that maintains cells into a "proliferative" and drug-sensitive state. In contrast, AhR activated by α-ligands (canonical ligands) promote a dedifferentiation state and a BRAFi resistance program. These latter cells are probably involved in the relapse. Our results are consistent with recent studies indicating that constitutive and chronic activation of AhR promotes aggressive tumor behavior[37,38] and tumorigenesis in vivo[39,40]. Indeed, AhR is now reported to have pro- or anti-tumor activity according to cell state[41–45]. Activation of AhR has also been associated with

the alteration of many cell differentiation processes[46] and melanoma cell dormancy[47,48]. Since the chronic exposure to α-ligands can switch BRAFi sensitive cells into persister/resistant cells, it is important to keep in mind that chronic activation of AhR occurs in response to many environmental factors, such as UV exposure[49] and pollutants[50], all known to promote cancer. Our data also highlight the potential of AhR antagonists as sensitizers of melanoma-targeted therapy. Together, these data underscore the importance of further studying the role of AhR-signaling in the context of cancer biology as a putative therapeutic target[41,51,52].

## Methods

**Reagents.** The AhR ligands were obtained from: 2,3,7,8-TCDD (TCDD) (Sigma Aldrich, 48599), FICZ (Sigma Aldrich, SML1489), Vemurafenib (Vem, PLX4032) (Selleckchem, RG7204), Dabrafenib (Dab) (Santa Cruz Biotechnology, SC364477), PLX7904 (MedChem Express, HY-18997), PLX8394 (MedChem Express, HY-18972), CH-223191 (Selleckchem, S7711), L-Kynurenine (Sigma Aldrich, K8625), Resveratrol (RSV) (Selleckchem, S1396), StemRegenin 1 (SR1) (Selleckchem, S2858), and Benzo(a)pyrene (B(a)P) (Sigma Aldrich, B1760).

**Plasmids.** The pGL3-XRE3-FL construct containing three XRE sequences from *CYP1A1* gene has been described previously[53]. Luciferase reporter plasmids (pOCA2-pGL4 luciferase) containing proximal promoter region −500b (IDT, Leuven, Belgium) was cloned into the pGL4.10 (Promega, USA) using Gibson Assembly® Master Mix following manufacturer's recommendations (NEB, UK). Plasmid encoding the MEK1 constitutive kinase form was described by ref. [54].

**Cell lines and culture conditions.** The mammary MCF7 epithelial cells were cultured in humidified air (37 °C, 5% $CO_2$) in Dulbecco's modified Eagle's medium with 4500 mg/l D-glucose, 110 mg/l sodium pyruvate, supplemented with 10% fetal bovine serum (PAA cell culture company) and 1% penicillin–streptomycin antibiotics (Gibco, Invitrogen). The melanoma cell lines (501Mel, MM001, SKmel28, MM074, Mel624) were grown in humidified air (37 °C, 5% $CO_2$) in RPMI-1640 medium (Gibco BRL, Invitrogen, Paisley, UK) supplemented with 10% fetal bovine serum (PAA cell culture company) and 1% penicillin–streptomycin antibiotics (Gibco, Invitrogen). Mel624 cells were obtained from G. Lizee's lab at the University of Texas MD Anderson Cancer Center, Houston, TX. MM001 and MM074 (S+R) cells were obtained from G. Ghanem's lab at the Institut Jules Bordet, Université Libre de Bruxelles, Brussels, Belgium. SKMel28 (S+R) cell was obtained from J.C. Marine's lab at VIB Center for Cancer Biology, VIB, Leuven, Belgium. 501Mel cells (S) were obtained from ATCC and 501Mel BRAFi resistant cells (R) have been obtained after 3 months treatment with Vem (1 μM every 2 days). All cell lines have been routinely tested for mycoplasma contamination.

**Molecular modeling.** Docking experiments were performed with AutoDock4.2; free open tool, http://autodock.scripps.edu[55]. A multiple alignment between the sequences of PAS-B mAhR (residues 278–384) and PAS-B HIF-2α was generated

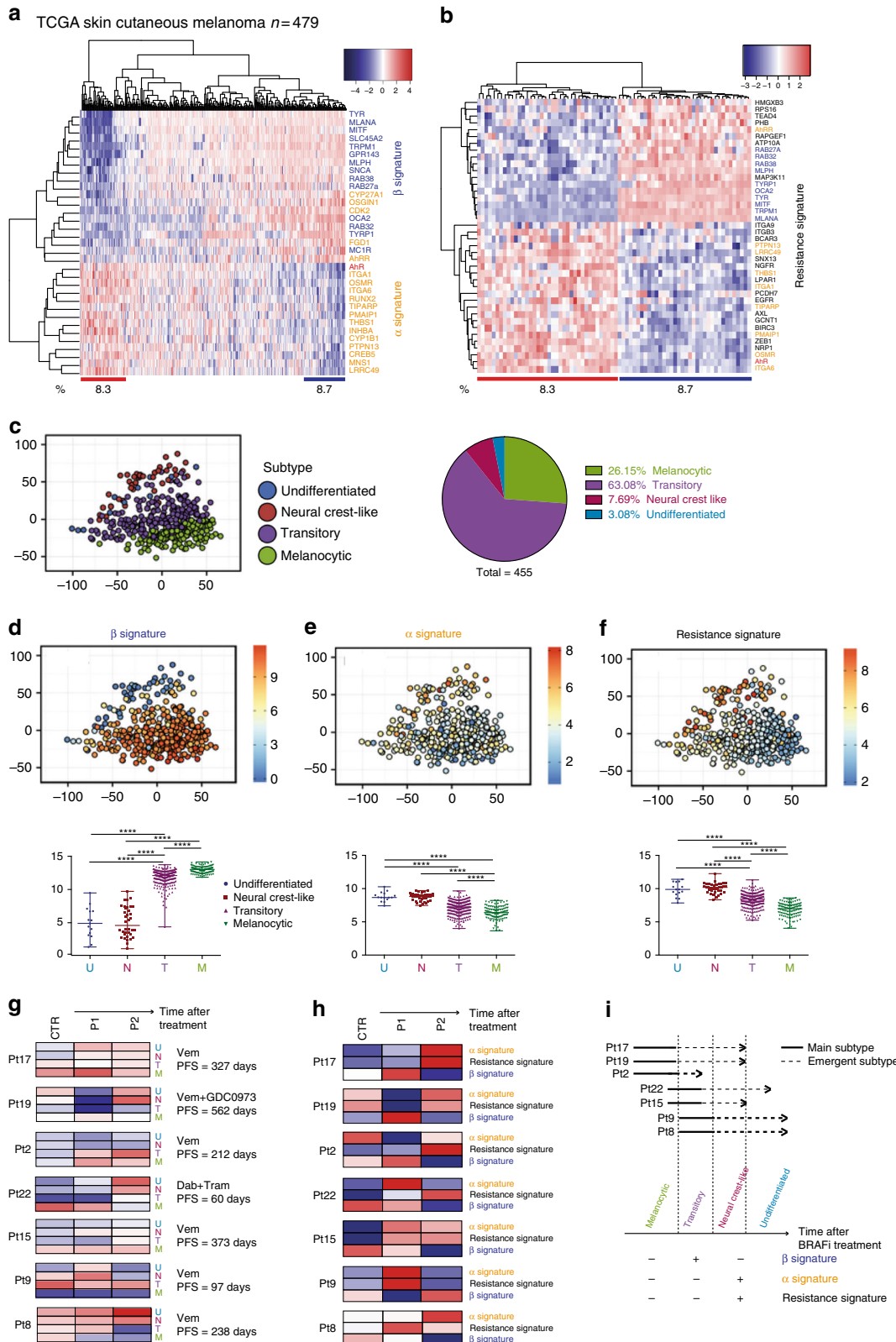

according to the sequence alignment suggested by Pandini et al.[56]. The homology model of PAS-B mAhR was constructed using the crystal structures of the heterodimer complex of PAS-B HIF-2α (pdb code: 3f1p, 3f1o, 3f1n) and Prime v.2.1. Docking experiments were carried out between AhR PAS-B model and different AhR ligands and BRAFi chemical structures recovered in ZINC database (zinc.docking.org).

**Cell density evaluation**. Cell density was assessed using a methylene blue colorimetric assay[57]. Briefly, cells were fixed for at least 30 min in 95% ethanol. Following ethanol removal, the fixed cells were dried and stained for 30 min with 1% methylene blue dye in borate buffer. After four washes with tap water, 100 μl of 0.1 N HCl were added to each well. Plates were next analyzed with a spectrophotometer at 620 nm.

**Fig. 3** In patients' tumors, AhR signatures correlate with cell-dedifferentiation states and resistance to BRAFi. **a** Expression heatmap depicting mRNA expression of individual genes for pigmentation signature (blue, β signature) and AhR target genes (orange, α signature) in non-treated melanoma patient dataset from TCGA (SKCM, $n = 459$)[29]. **b** Expression heatmap depicting mRNA expression of individual genes for BRAFi resistance genes in non-treated melanoma patients dataset from TCGA (SKCM, $n = 459$) with high level ($n = 40$) or low level of expression ($n = 34$) for AhR. **c** PCA of TCGA datasets obtained by the cluster prediction assignment (melanoma dedifferentiation signature resource from the Graber lab: http://systems.crump.ucla.edu/dediff/)[10] and pie-chart representation of the melanoma dedifferentiation subtypes. **d-f** MLANA, PMAIP1, and AXL expression PCA color maps illustrating, respectively, β- (**d**), α- (**e**), and resistance signatures (**f**) from the TCGA dataset (top). Boxplots of selected β-, α-, and resistance genes (as described above) in different untreated melanoma patients' biopsies from TCGA (U, undifferentiated; N, neural crest-like; T, transitory; M, melanocytic) (bottom). Number in each group: U = 14, N = 35, T = 287, M = 118. Whiskers reflect median of expression with range. One-way ANOVA and Tukey's test: ***$p < 0.001$, ****$p < 0.0001$. **g** Expression heatmap for melanoma differentiated state signatures in BRAFi-treated melanoma. **h** Expression heatmap for average β-, α-, and resistance signatures in BRAFi-treated single-drug (i.e., BRAFi) or double-drug (i.e., BRAFi + MEKi) melanoma patients during resistance acquiring (RNAseq dataset GEO, GSE65185)[13]. Clinical data are available from supplemental table S1 from Hugo et al.[13]. **i** Schematic representation of temporal transcriptional regulation of different signatures in a melanoma patient treated with BRAFi

**Picture from patients**. Picture of Vem treated melanoma patient was obtained from Dr. Lise Boussemart at the Medical Department of Dermatology, CHU Rennes, France after informed consent.

**AhR ligand binding assay**. Ligand binding was measured as the presence/absence of ligand-dependent displacement of [$^3$H]TCDD from the hepatic cytosolic guinea pig AhR by the hydroxyapatite assay. The experiments were performed essentially as previously described[58].

**Gel retardation assay**. Experiments have been done as previously described[59].

**AhR immunolocalization**. Experiments have been done as previously described[53].

**Ethoxyresorufin O-deethylase activity assay**. Ethoxyresorufin O-deethylase (EROD) activity, corresponding to the O-deethylation of ethoxyresorufin, and mainly supported by the CYP1A1 enzyme in living MCF7 cells, was measured as described previously[53].

**RNA interference**. Plasmids encoding shRNA targeting human AhR (TL320259, 29mer shRNA constructs in lentiviral GFP vector) were purchased from Origene, Rockville, MD. Lentiviral productions have been performed as recommended (http://tronolab.epfl.ch), using 293T cells, psPAX2, pVSVG, and shRNA. Infections were performed overnight in the presence of 8 µg of polybrene per ml. After infection, cells were maintained under selection in the presence of puromycin (Invivogen, San Diego, CA) and seeded in 96-well plates at 0.5 cells/well for single cell clonal expansion. Clones of interest were validated by Western blot analysis and RT-qPCR.

siRNA were purchased from Sigma-Genosys (St. Louis, MO, USA): siCTR, siOCA2#1 and #2, siARNT. Sequences for all shRNA or siRNA are available in Supplementary Table 2.

**CRISPR/Cas9 experiment**. AhR knock out has been performed using CRISPR/Cas9 methodology. Guide sequence targeting AhR (available in Supplementary Table 3) (Sigma-Genosys, St. Louis, MO, USA) has been cloned into the GeneArt CRISPR Nuclease vector according to the manufacturer's instructions (Life Technologies, Saint-Aubin, France). Next, vectors were transfected in 501Mel cells, and 2 days later cells were seeded in 96-well plates at 0.5 cells/well for single cell clonal expansion. Clones of interest were validated by DNA-sequencing, Western blot analysis, and RT-qPCR.

**Luciferase activity**. $2 \cdot 10^5$ MCF7 or 501Mel cells were cultured in 12-well plates and transfected with respectively the pGL3-XRE3-FL[53] and the pGL4-pOCA2-Luc constructs carrying the firefly luciferase. Transient transfection of cells was performed by the JetPRIME® transfection reagent according to the manufacturer's instructions (Polyplus transfection™, NY, USA). In brief, 50 µl of JetPRIME® Buffer containing 100 ng of firefly luciferase reporter plasmid was added per well, and 1 µl of JetPRIME® transfection reagent. After a 24-h period, cells were exposed to TCDD, Vemurafenib for a 6-h period. Luciferase assays were then performed with a Promega kit according to the manufacturer's instructions. Data were expressed in arbitrary units, relative to the value of luciferase activity levels found in DMSO-exposed cells, arbitrarily set at 1 arbitrary unit (a.u.). Firefly luciferase activity was normalized to protein content using Bicinchoninic Acid Kit from Sigma-Aldrich® and measured with using a luminometer (Centro XS$^3$ LB960, Berthold Technologies).

**RNA extraction and RT-qPCR expression**. Experiments have been done as previously described[60]. Primers used for RT-qPCR experiments are available in Supplementary Table 3.

**Chromatin immunoprecipitation assay**. ChIP assays, using $2 \times 10^6$ 501Mel cells or 501 KD AhR (as negative control), were performed as previously described[61], with specific adaptations. The cells were cross-linked (1% final concentration formaldehyde for 10 min), washed twice and collected in 1 ml cold PBS. Cells were lysed and the samples were then sonicated for DNA fragmentation (Sonifier Cell Disruptor, Branson) in 1 ml lysis buffer (10 mM EDTA, 50 mM Tris–HCl (pH 8.0), 1% SDS, 0.5% Empigen BB) and diluted 2.5-fold in IP buffer (2 mM EDTA, 150 mM NaCl, 20 mM Tris–HCl (pH 8.1), 0.1% Triton X-100). This fraction was subjected to immunoprecipitation overnight with 4 µg of the appropriate antibody (AhR, H211, Santa Cruz). These samples were then incubated for O/N at 4 °C with 20 µl of Protein G Dynabeads™ (Invitrogen). Precipitates were washed several times, cross-linking reversed and DNA purified using a Nucleospin Extract II kit (Macherey Nagel).

qPCR analyses were carried out with primers spanning target genes proximal promoters (sequences on Supplementary Table 3). For qPCR analysis, fold enrichment was determined using the ΔΔCt method: fold enrichment = $2^{-(\Delta ct1 - \Delta Ct2)}$, where ΔCt1 is the ChIP of interest and ΔCt2 the control ChIP.

**Western blot**. Harvested cells were solubilized as previously described[57]. Protein samples were denatured at 95 °C, resolved by SDS-PAGE and transferred onto Hybond™-C Extra nitrocellulose membranes (Amersham Biosciences, Bucks, UK). Membranes were probed with appropriated antibodies and signals were detected using the Fujifilm LAS-3000 Imager (Fuji Photo Film, Tokyo, Japan). Primary antibodies were: anti-Phospho-ERK-1/2 (9101S), MEK-1/2 (D1A5) (Cell Signalling Technology, Boston, USA), ERK-1/2 (K23), AhR (H211), Hsc70 (B6) (Santa Cruz Biotechnology, Santa Cruz, CA). Horseradish-peroxidase-conjugated secondary antibodies were purchased from (Jackson ImmunoResearch (Suffolk, UK)) (1:1000).

**Proximity ligation assay**. The proximity ligation assay was applied in order to visualize AhR/ARNT complexes in MCF7 cells. The cells, grown on glass coverslips, were fixed with 4% PFA in 0.1 M phosphate buffer (15735-60S, Electron Microscopy Sciences) for 15 min at RT and PLA was performed using the kit ((DUO92007) Duolink® in Situ Detection Reagent Orange, (DUO92001) Duolink® in Situ PLA® Probe Anti-Mouse PLUS, (DUO92005) Duolink® in Situ PLA® Probe Anti-Rabbit MINUS, SIGMA) according to the manufacturer's protocol. After blocking, the reaction with primary antibodies, mouse anti-AhR (C20, 1/100) and rabbit anti-ARNT (1C12, 1/100). Following the ligation and amplification steps, the coverslips were immobilized on the microscopic slides with the mounting medium containing DAPI. In control experiment, the ligation step was omitted. Imaging analysis was carried out using a delta vision system (Applied Precision). Number of foci was quantified in at least 30 cells.

**RNA-Seq**. Total RNAs was extracted from 501Mel cells treated for 48 h respectively with DMSO, Vemurafenib (1 µM), Resveratrol (5 µM), Resveratrol (5 µM) + Vemurafenib (1 µM), TCDD (10 nM), Resveratrol (5 µM) + TCDD (10 nM) using the miRVana kit (Thermo Fisher Scientific). Libraries were generated from 500 ng of total RNAs using Truseq Stranded mRNA kit (Illumina). Libraries were then quantified with KAPA library quantification kit (Kapa Biosystems) and pooled. 0.5 nM of this pool were loaded on a high output flowcell and sequenced on a NextSeq500 platform (Illumina) with $2 \times 75$ bp paired-end chemistry within two runs. Reads were aligned to the human genome release hg19 using STAR v2.4.0a with default parameters. Quantification of genes was then performed using featureCounts release subread-1.5.0-p3-Linux-x86_64 with "--primary -g gene_name -p -s 1" options. Quality control of RNA-Seq count data was assessed using in-house R scripts. Normalization and statistical analysis were performed using the Bioconductor package DESeq2. p-Values were adjusted for multiple testing using the Benjamini–Hochberg procedure, which controls the false discovery rate (FDR). Differentially expressed genes were selected based on an adjusted p-value below 0.05. The RNA-Seq data presented in this article have been submitted to the Gene

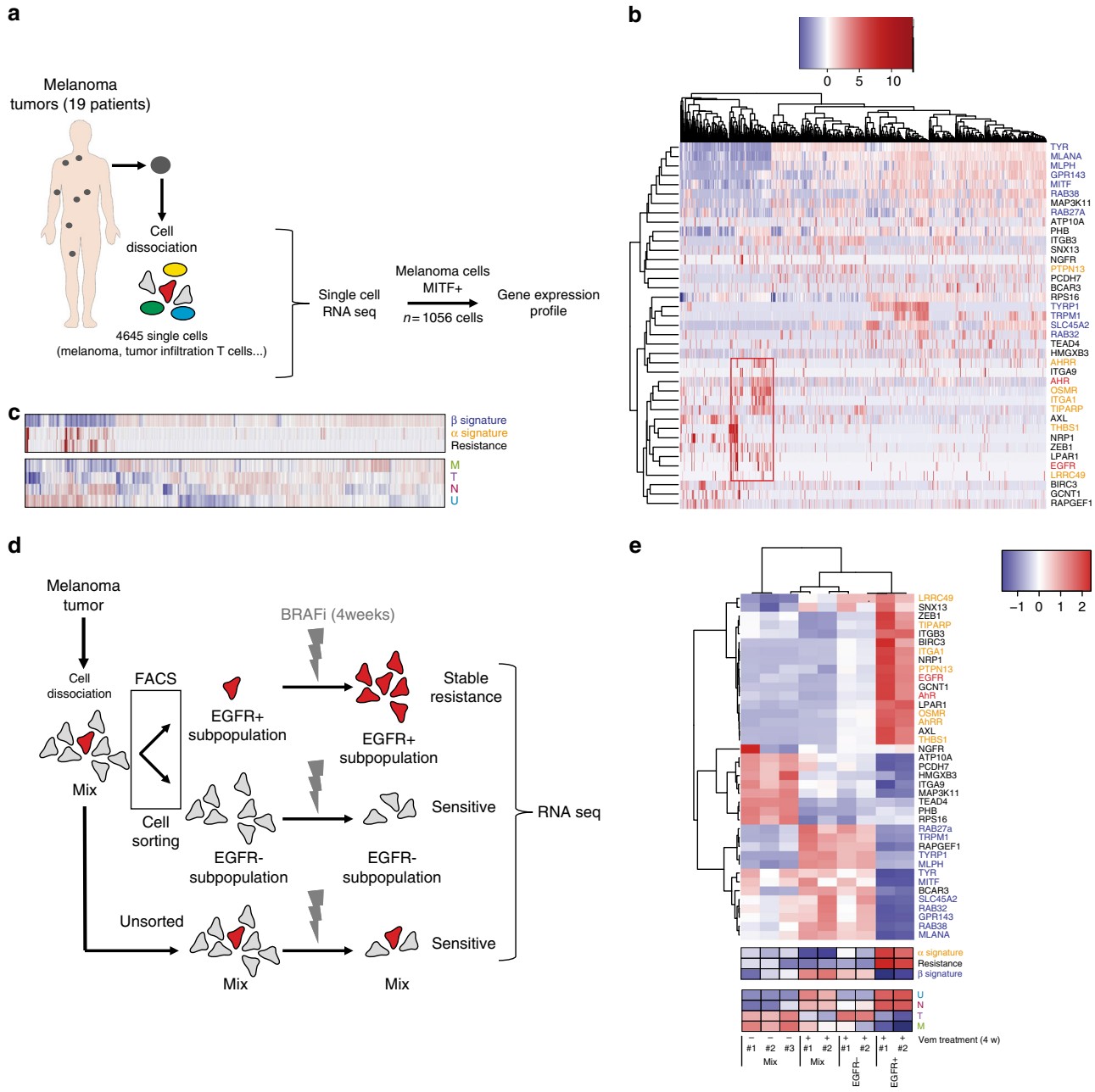

**Fig. 4** Role of AhR in melanoma cell resistance: single cell analyses. **a** Schematic representation of analysis of data from single cell analysis of BRAFi-resistant melanoma (19 patients). Cells from tumors of the different patients have been dissociated and individually sequenced for RNA expression (4650 cells) (RNAseq dataset, GEO, GSE72056[33]). One filter has been included to only focus on melanoma cells expressing the *MITF* gene. **b** Expression Heatmap for genes corresponding to α- and β-signatures and depicting mRNA expression of individual genes for BRAFi resistance genes using data from single cell analysis of BRAFi-resistant melanoma (19 patients) (1056 cells). **c** Expression heatmap for average β-, α-, and resistance signatures (top) and melanoma differentiated state signatures using data from single cell analysis of BRAFi-resistant melanoma. **d** Schematic representation of analysis of data using an RNA-Seq dataset obtained from BRAFi-treated melanoma cells (from 2 patients) (from supplemented information[16]). Cell sorting has been performed on dissociated tumors to isolate EGFR-negative or -positive cells that are able to resist BRAFis and to generate colonies after long-term treatment. **e** Expression heatmap for genes corresponding to individual genes (top) of α- and β-signatures and depicting mRNA expression for BRAFi resistance, average signatures (middle) and differentiated state signatures (bottom) using an RNA-Seq dataset obtained from BRAFi-treated melanoma cells[16]. The human silhouettes have been adapted (change of color background) from Servier Medical Art, licensed under a CC BY 3.0 FR [https://smart.servier.com/smart_image/shape-29/]

Expression Omnibus database (http://www.ncbi.nlm.nih.gov/geo/) under the accession number GSE104869.

**Patient-derived xenografts (PDXs).** After approval by the University Hospital KU Leuven Medical Ethical Committee (S54185) and written informed consent from the patient, PDX model MEL006 was established from an in-transit

metastasis resected as part of standard-of-care melanoma treatment at the University Hospital KU Leuven. The procedures involving mice were performed in accordance with the guidelines of the IACUC and KU Leuven and carried out within the context of approved project applications P147/2012, P038/2015, and P098/2015. Fresh tumor tissue was collected in transport medium (RPMI1640 medium supplemented with penicillin/streptomycin and amphotericin B). Tumor fragments were subsequently rinsed in phosphate-buffered saline

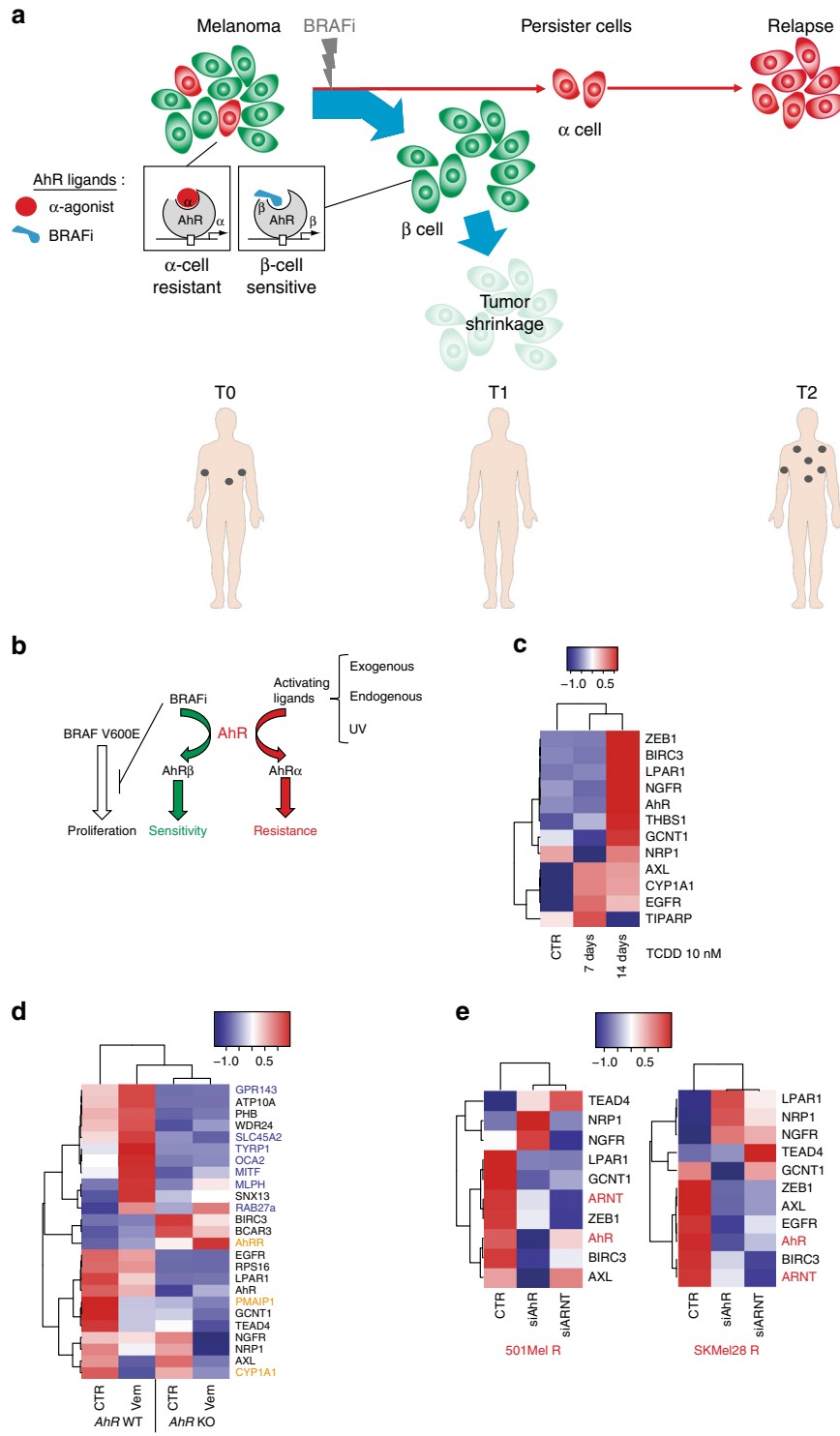

**Fig. 5** Long-term canonical activation of AhR drives melanoma resistance to BRAFi. **a** Graphical representation of AhR function controlling melanoma cell sensitivity or resistance during BRAFi treatment. A high level of heterogeneity is observed among melanomas with a high proportion of highly differentiated and β-cells sensitive to BRAFi (induction of pigmentation by AhR: β signature) and a weak number of undifferentiated and α-cells resistant to BRAFi (induction of α signature and resistance genes). These persister cells constitute a cell reservoir leading to melanoma relapse. **b** Graphical model of AhR activation by BRAFi and α-ligands, with α-ligands dictating melanoma resistance. **c** Expression heatmap for resistant genes in 501Mel cells treated for 7–14 days with TCDD (10 nM). **d** 501 melanoma cells (501Mel) were pre-treated daily or not for 2 weeks with TCDD (10 nM) and treated 4 days with increasing concentrations of Vem in order to establish cell density measurements and calculate IC50 (sensitivity to Vem). Values, calculated with GraphPad (PRISM6.0®), represent the IC50 of Vem for control cells (without TCDD pre-treatment) or after 2 weeks of TCDD. **e** Expression heatmap for β-, α-, and resistance genes in 501Mel cells invalidated or not for AhR by CRISPR/Cas9 before or 48 h after treatment with Vem (1 μM). **f** Expression Heatmap for β-, α-, and resistance genes in 501Mel and SKMEL28 (R) cells knocked-down for *AhR* or *ARNT* using siRNA. The human silhouettes have been adapted (change of color background) from Servier Medical Art, licensed under a CC BY 3.0 FR [https://smart.servier.com/smart_image/shape-29/]

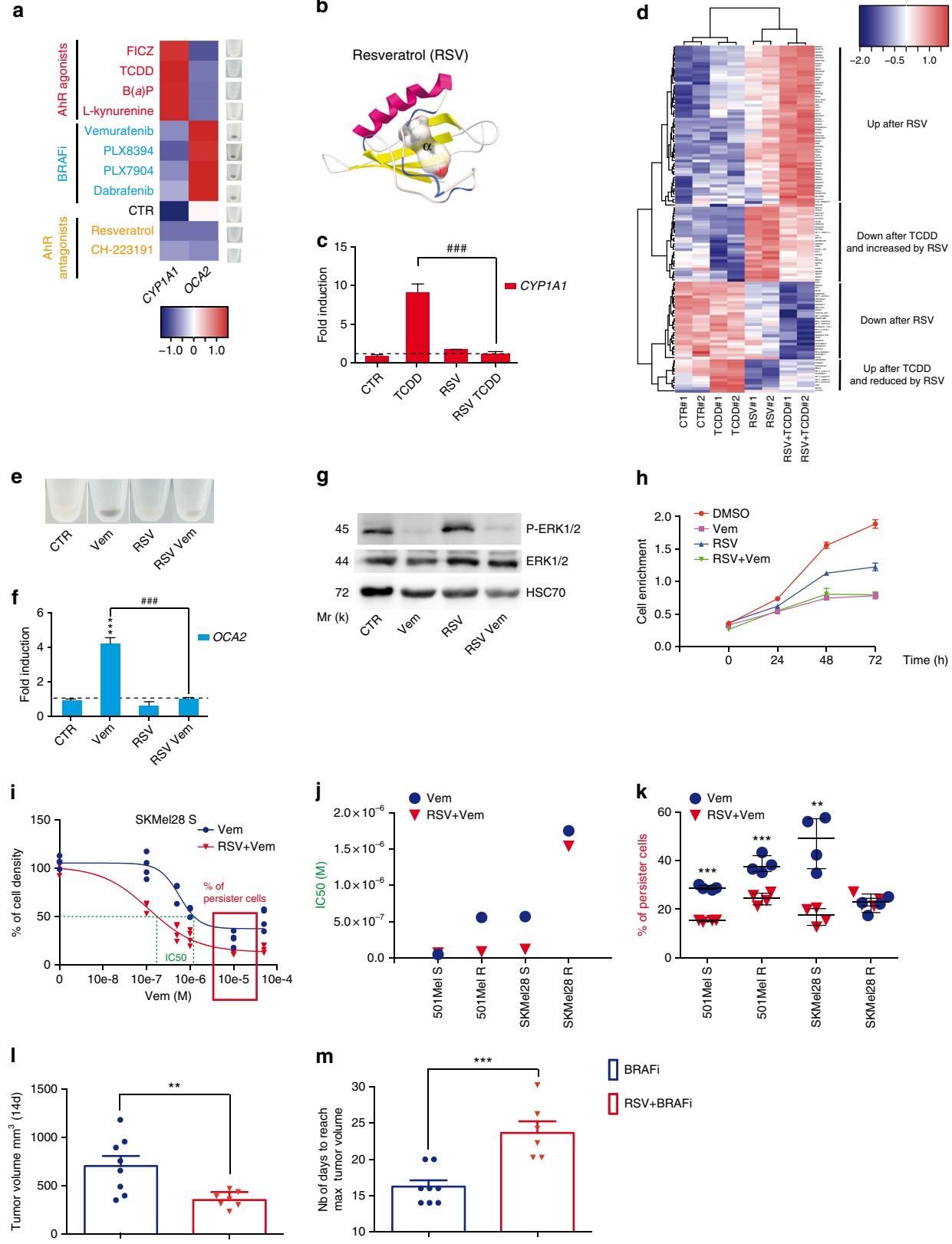

supplemented with penicillin/streptomycin and amphotericin B and cut into small pieces of approximately $3 \times 3 \times 3$ mm$^3$. Tumor pieces were implanted in the interscapular fat pad of female SCID-beige mice (Taconic). After reaching generation 4 (F4), tumor fragments were implanted in the interscapular fat pad of female NMRI nude mice (Taconic). Ketamine, medetomidine, and buprenorphine were used for anesthesia.

**Pharmacologic treatment of mice.** Mice with tumors reaching 200–300 mm$^3$ were treated via daily oral gavage. Dabrafenib (Biorbyt) and/or Resveratrol (Selleckchem) were dissolved in DMSO at a concentration of 30 and 40 mg/ml respectively, aliquoted and stored at −80 °C. Each day a new aliquot was diluted 1:10 with phosphate-buffered saline and mice were treated with a dose of 30 and 40 mg/kg for Dabrafenib and Resveratrol, respectively. Tumor volume was

**Fig. 6** Therapeutic opportunity to limit BRAFi resistance. **a** Heatmap depicting the effects of different AhR ligands on *OCA2* and *CYP1A1* mRNA in 501Mel cells and pigmentation (48 h). Three groups: exo- and endo-gene ligands (TCDD, B(*a*)P and FICZ, L-Kynurenine, respectively), BRAFi and AhR antagonists (Resveratrol and CH-223191). **b** Binding model of the antagonist Resveratrol (RSV) to the PAS B of AHR. RSV is predicted to bind to the α-pocket. Free binding energy is reported in Supplementary Table 1. **c** RSV prevents *CYP1A1* mRNA induction (48 h) by TCDD. 501Mel cells were pre-treated with 5 μM RSV 2 h before 10 nM TCDD. **d** Gene expression profile of 501Mel cells exposed to vehicle, TCDD (10 nM), RSV (5 μM), or RSV (5 μM) + TCDD (10 nM) ($n = 2$) for 48 h. Heatmap focused on differentially expressed genes as a function of treatment. **e–g** 501Mel cells were treated with Vem (1 μM) alone or in combination with RSV (5 μM) for 48 h for pigmentation analyses (**e**), *OCA2* mRNA expression levels (**f**), and phospho-ERK and ERK total detection by Western blotting (**g**). **h** 501Mel cells were pretreated for 2 h with RSV (5 μM) before Vem addition (1 μM). 501Mel cell density was evaluated by methylene blue staining followed by quantification at 620 nm ($n = 2$). **i–k** Two pairs of BRAFi-sensitive (S) and -resistant (R) melanoma cells (501Mel and SKMel28) were pre-treated or not for 1 week with RSV (1 μM, every 2 days) before treatment with Vem in order to establish Vem IC50 3 days after BRAFi treatment. Values, calculated with GraphPad PRISM (**i**), represent IC50 of Vem for control cells (without RSV pre-treatment) or after 1 week of RSV (**j**). % of BRAFi-persister cell values correspond to the percentage of residual cells following 3 days of Vem (5 μM) treatment in comparison to melanoma cells without RSV treatment (**k**). **l** PDX tumor volumes 14 days after daily treatment with Dabrafenib (30 mg/kg) ($n = 8$) or in combination with RSV (40 mg/kg) ($n = 7$). **m** Number of days to reach max tumor volume (endpoint: >800 mm$^3$). Values correspond to the mean ± sem. Two-tailed unpaired $t$ test for the different treatments was performed: **$p < 0.01$

monitored with a caliper and calculated using the following formula: $V = (\pi/6)$ *length*width*height.

**Data mining**. Analysis of TCGA/SKCM data was performed using OncoLnc portal [http://www.oncolnc.org][62]. The raw data count matrix composed of 479 samples; from SKCM melanoma cohort was downloaded from the OncoLnc for the pigmentation genes ($n = 14$)[12], the AhR target genes ($n = 19$)[63], or the BRAFi resistant genes ($n = 19$)[17]. Expression heatmap of differentially expressed genes between samples was obtained on log2 fold change using heatmap3 package in R/ Bioconductor. Cluster-specific genes rankings were obtained by contrasting the samples with the rest of the samples. Cell density curves for the available melanoma cell lines were established using GraphPad PRISM 6.0® in order to establish IC50 dependently to the different treatments.

The raw data count matrix from RNA seq data were obtained in GEO database for previous experiments on melanoma cell lines Cancer Cell Line Encyclopedia[29] GSE36134 (sensitive or resistant to PLX470) (IC50 values for PLX4720 were obtained from the Supplementary Table 7 of ref. [29]); on BRAFi or BRAFi + MEKi resistant cell lines GSE75299[32] and GSE80829[10], on BRAFi treated melanoma patients GSE65185[13]; on primary melanoma cell lines (proliferative or invasive) GSE60664[15], on single cell analysis of BRAFi resistant melanoma (4650 seq) GSE72056[33], and on Vem-resistant melanoma (EGFR pos)[16]. Expression heatmap of differentially expressed genes (pigmentation, AhR, or resistant genes) between samples was obtained on log2 fold change using heatmap3 package in R/ Bioconductor. PCA color maps illustrating expression of genes in melanoma cell lines dataset have been obtained by the cluster prediction assignment (from Melanoma dedifferentiation signature resource from Graeber's lab: [http://systems. crump.ucla.edu/dediff/]).

**Statistics**. Data are presented as mean ± s.d. unless otherwise specified, and differences were considered significant at a $p$ value of less than 0.05. Comparisons between groups normalized to a control were carried out by a two-tailed $t$-test with the Holm–Sidak's multiple comparisons test when more than two groups are compared to the same control condition. OS was estimated using the Kaplan–Meier method. Univariate analysis using the Cox regression model was done to estimate hazard ratios (HR) and 95% confidence intervals (CI). All statistical analyses were performed using Prism 6 software (GraphPad, La Jolla, CA, USA).

## Data availability

The datasets generated during and/or analyzed during the current study are available from the corresponding author on reasonable request.

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

## Acknowledgements

The authors thank the Gene Expression and Oncogenesis team for helpful discussions; the CNRS UMR6290; the Rennes FHU CAMIn team. The authors thank Odessa Van Goethem from VIB Center for Cancer Biology, VIB, Leuven, Belgium for PDX experiment in mice. The authors also thank Robyn Tolhurst from MND and Neurodegenerative Diseases Research Center, Sydney and Nicolas Nottet and Bernard Mari from Institute of Molecular and Cellular Pharmacology UMR7275 CNRS, Nice. The authors acknowledge the SFR Biosit core facilities of Rennes University with the cell imaging ImPACcell (Rémy Le Guevel), Microscopy Rennes Imaging Center (MRIC) platforms. This study received financial support from the following: AVIESAN plan Cancer (ENV201308 and ENV201603), Fondation ARC (No. PGA1*20160203868), Ligue National Contre le Cancer (LNCC) Départements du Grand-Ouest; Association "Vaincre le Cancer", FHU CAMIn-CHU Rennes, Région Bretagne; University of Rennes 1; CNRS, Inserm. N.T. is a recipient of a doctoral fellowship from ministère de la Recherche. A.G. is a recipient of Ligue Contre le Cancer (Grand-Ouest).

## Author contributions

Conceptualization: S.C., D.G., and M.-D.G. Methodology: S.C., N.T., N.M., H.M.L., A.G., L.Ba., A.P., A.S., A.R., E.D., K.T. and D.G. Investigation: S.C., D.G., N.T., N.M., H.M.L., A.G., A.P., L.Ba., A.S., and A.B. Formal analysis: S.C., N.T., N.M., H.M.L., and D.G. Writing—original draft: S.C., D.G., and M.-D.G. Funding acquisition: S.C., D.G., and M.-D.G. Resources: L.Bo., F.R., G.J.G., J.-C.M. and M.S.D. Supervision: S.C., D.G., and M.-D.G.
