## [Peer Review File · Nature Communications]

Reviewers' comments:

Reviewer #1 (Remarks to the Author):

This is an intriguing but ultimately confusing study that aims to address the role of the Aryl hydrocarbon receptor in BRAF inhibitor resistance in melanoma. The authors show that a number of BRAF inhibitors activate the aryl hydrocarbon receptors in melanoma cells and that this can maintain melanoma cells in a drug sensitive state. It is further shown that BRAF inhibitors do not activate the Aryl hydrocarbon receptor in a classical manner and it is distinct from the natural ligands of this receptor – which instead activate a resistance program. The authors then show that use of an AhR antagonist can prevent the enrichment of resistant cells and that these reduce the percentage of persister cells on BRAF inhibitor therapy. Overall this is an intriguing story with some significant problems; the mechanism of resistance mediated through AhR is not really addressed and this new data is difficult to integrate with our current understand of BRAF inhibitor resistance. The lack of in vivo studies also diminished the impact of this study.

Major points

1. The authors don't really address how the Aryl hydrocarbon receptor mediates resistance. Does it directly alter resistance associated transcription factors in melanoma such as c-JUN or TGF/BMP transcription factors? Is the resistance associated with increased pERK recovery or RTK signaling or some other mechanism? All that is shown is that Aryl hydrocarbon receptor signaling is associated with a resistance associated gene signature. No mechanistic link is provided between the two. This is a major failing of the study.
2. Figure 1i showing multiple nevi on the back of a patient is just anecdotal and it is completely unscored. It adds very little to the story.
3. There are problems with the experiments showing that the vemurafenib induced B-signatures are not mediated through paradoxical ERK activation. 501Mel is a BRAF mutant melanoma cell line, why would this experience paradoxical ERK activation. It would make more sense to repeat these experiments in NRAS mutant melanoma that show paradoxical ERK activation following treatment with vemurafenib.
4. Most of the key experiments are just performed on two melanoma cell lines. Melanoma is very heterogeneous tumor; some of the key findings need to be replicated across a panel of BRAF mutant melanoma cell lines.
5. There are no in vivo studies showing the utility of targeting the aryl hydrocarbon receptor and that it can prevent the onset of resistance.
6. What is the effect of the BRAF-MEK inhibitor combination on Aryl hydrocarbon receptor signaling? Single agent BRAF inhibition is not routinely used any more and has been superseded by the combination. This would improve the translational impact of these studies.
7. Extended Figure 6a has no scale. What does the green shading mean?
8. Minor point: often the English is not good. The paper should be reviewed and corrected by a native English speaker.

Reviewer #2 (Remarks to the Author):

BRAF inhibitors are important for treatment of a large subset of cutaneous melanoma patients; however, the response to these agents is limited due to development of drug resistance. This manuscript identified the Ah receptor as a key mediator of BRAFi-resistant genes and shows that resveratrol, a putative Ah receptor antagonist, inhibits BRAFi-resistant genes, confirming a potential role for Ah receptor ligands in combination therapy with BRAF inhibitors for treating cutaneous melanoma. This is an interesting and novel observation; however, the authors need to address the following issues.

1. The overall description of Vem in the first part of Figure 1 could also be interpreted as the action

of a novel antagonist of the canonical AhR signaling pathway. The authors should show the combined effects of TCDD+Vem on EROD/luciferase activity, CYP1A1 mRNA levels, DRE binding, and related activities.

2. The ChIP assay in Extended Figure 2e is somewhat puzzling and requires further explanation. Does this represent constitutive or Vem-induced ligand binding and are similar results observed for both TCDD and Vem? The GCGTG core binding sites and flanking region traditionally bind AhR:Arnt and AhR binding alone is usually minimal to weak. It would be worth looking at a few AhR coactivators to see if Vem but not TCDD induces their recruitment. Identification of a new partner for the AhR is of prime interest but may be beyond the scope of this manuscript. The effects of CH223191 on AhR binding (+/- Vem) would also be interesting and complement results in extended Figure 3.

3. The results of the genomic and functional studies are compelling; however, the authors have ignored the prior studies in this area which show different results. For example,

(i) Leflunomide (an AhR-active pharmaceutical) inhibits melanoma cell growth (PLOS ONE 7, e40926, 2012).

(ii) The AhR has tumor suppressor activity with respect to melanoma growth and metastasis (Carcinogenesis 34, 2683, 2013).

(iii) Other studies show that the AhR and/or its ligands enhance melanoma growth/invasion (Nature 483, 603, 2012; Tox. Appl. Pharmacol. 210, 212, 206). All of these data have to be cited and rationalized.

4. The authors state the following conclusion: "These results are entirely consistent with recent studies indicating that constitutive and chronic activation of AhR promotes aggressive tumor behavior (31,32) and tumorigenesis in vivo (33, 34)." This statement is incorrect since the AhR and/or its ligand can act as tumor suppressors (colon, breast, pancreatic, ...) or tumor promoters (lung, glioblastoma, ...). The authors should read recent reviews on the AhR and cancer which clearly show these tissue/tumor-specific differences.

NCOMMS-18-02506
Answers to reviewers

We first would like to thank the reviewers for their interest in the present work and for their comments improving the manuscript. The reviewers had several concerns that we address in this letter.

In the revised version, we have widely implemented the data regarding the role of AhR in the emergence of resistant cells at the molecular level using *in vitro* and *in vivo* experiments and publicly available data from recent publications ¹⁻⁶

To ease the reading, we have indicated the modifications that are included in the manuscript.

Reviewer 1

This is an intriguing but ultimately confusing study that aims to address the role of the Aryl hydrocarbon receptor in BRAF inhibitor resistance in melanoma. The authors show that a number of BRAF inhibitors activate the aryl hydrocarbon receptors in melanoma cells and that this can maintain melanoma cells in a drug sensitive state. It is further shown that BRAF inhibitors do not activate the Aryl hydrocarbon receptor in a classical manner and it is distinct from the natural ligands of this receptor – which instead activate a resistance program. The authors then show that use of an AhR antagonist can prevent the enrichment of resistant cells and that these reduce the percentage of persister cells on BRAF inhibitor therapy. Overall this is an intriguing story with some significant problems; the mechanism of resistance mediated through AhR is not really addressed and this new data is difficult to integrate with our current understand of BRAF inhibitor resistance. The lack of *in vivo* studies also diminished the impact of this study.

Major points

1. The authors don't really address how the Aryl hydrocarbon receptor mediates resistance. Does it directly alter resistance associated transcription factors in melanoma such as c-JUN or TGF/BMP transcription factors? Is the resistance associated with increased pERK recovery or RTK signaling or some other mechanism? All that is shown is that Aryl hydrocarbon receptor signaling is associated with a resistance associated gene signature. No mechanistic link is provided between the two. This is a major failing of the study.

We thank the reviewer for this critical comment. Indeed, it is now becoming clear that drug resistance does not only occur through the acquisition of specific gene alteration (i.e. RAS, MEK, COT...). c-JUN, TGFβ but also PI3K activation have been shown to mediate melanoma resistance programs. Also during the reviewing process of the present manuscript, two seminal papers ^{1,7} proposed that BRAFi resistance may be due to melanoma cells dedifferentiation, supporting the notion of genetic reprogramming as a potent mechanism of BRAFi-resistance.

The JC Marine Lab (co-author of the present paper) proposed in their publication "Towards minimal residual disease-directed therapy in melanoma" ⁷ that according to recent *in vitro* findings the emergence of these resistant cells is observed at a frequency much higher than would be expected due to mutational mechanisms ^{8,9}. They propose a non-mutually exclusive scenario where drug-tolerant phenotype is transiently acquired by a small proportion of cancer cells, through non-mutational mechanisms such as epigenetic and/or transcriptome reprogramming ^{8,9}.

In accordance with these recent findings, we identified here AhR as a central upstream mediator of BRAFi resistance. We provide evidence that selection of pre-existing resistant clone(s) displaying a sustained AhR activation (α -signature and resistance signature; Fig. 3a) contribute to minimal residual disease (MRD) and relapses (Fig. 3). We also demonstrate that emergence of resistant cells is primarily driven by adaptive, non-mutational events since differentiated and BRAFi-sensitive cells (β -signature) can be redirected towards an AhR-dependent resistant program (α -signature) using an AhR canonical agonist (Fig. 5c).

Cell state transition -drives at least by sustained AhR activation- contributes to emergence of BRAFi resistant cells and raise the possibility that selective inhibition of the master regulator governing the dedifferentiation may offer an effective approach for delaying or even preventing the development of resistance to melanoma targeted therapy. Consequently, we demonstrated *in vivo* (PDX model) that targeting AhR with antagonist is pertinent to delay or abrogate relapses in patients (Fig. 6i, see also comment N°5).

To deepen the understanding of how AhR mediates resistance, we showed that genes known to mediate resistance (AXL, TEAD4, NRP1 among others) and part of the resistant/ α -signature are direct targets of AhR. By ChIP-qPCR experiments performed in 501 Mel cells exposed to AhR canonical α -ligand (TCDD), we indeed showed that AhR binds to AXL, TIPARP, NRP1, TEAD4 promoters which contain XREs binding motifs (GCGTG) (Extended data Fig. 6). Importantly, and as mentioned above, sustained activation of AhR by its canonical ligand during 14 to 21 days, leads to the induction of the expression of these resistant genes (Fig. 5c).

Finally, melanoma patients (Pt19 & Pt22) under BRAFi/MEKi double blockade experienced an β - to α -signature transition supporting the notion that AhR-resistant program is MEK-independent (Fig. 3g-l; and comment N°4).

Together, these experiments, in addition to the genetic and pharmacology loss of AhR function, clarify the initial link associating AhR to α - and resistance signature observed in response to BRAFi.

2. Figure 1i showing multiple nevi on the back of a patient is just anecdotal and it is completely unscored. It adds very little to the story.

We thank the reviewer for this wise comment and have moved this illustration to the Extended data Section. It is now part of the Extended data Fig. 2.

3. There are problems with the experiments showing that the vemurafenib induced B-signatures are not mediated through paradoxical ERK activation. 501Mel is a BRAF mutant melanoma cell line, why would this experience paradoxical ERK activation. It would make more sense to repeat these experiments in NRAS mutant melanoma that show paradoxical ERK activation following treatment with vemurafenib.

We agree with the reviewer's comment. The 501Mel cell line is probably not the best cell line for such demonstration. However, we found that BRAFi-induced β -signature is observed in BRAF and NRAS mutated cell lines showing that BRAFi-induced β -signature is independent of BRAF/NRAS status (Fig. 2h). Moreover, Dabrafenib and last generation of BRAF V600E inhibitors known as "Paradox breakers" (PLX8394 and PLX7904) give similar results than Vem (Extended data Fig. 4 and Extended data Fig. 8a). Finally, we show that Vem-induced β -signature is maintained in the presence of MEKi (Extended data Fig. 4a-c). Altogether, these results strongly support that BRAFi-induced β -signature is not linked to paradoxical ERK activation (Extended Data Figure 3).

To clarify this point we have rephrased this section (L87-L90).

4. Most of the key experiments are just performed on two melanoma cell lines. Melanoma is very heterogeneous tumor; some of the key findings need to be replicated across a panel of BRAF mutant melanoma cell lines.

We agree with the reviewer, we have mainly been using two melanoma cell lines (501Mel and SKMel28) as working models to investigate the role of AhR in cell fate decision and to establish the three signatures reflecting the sensitivity/resistance to BRAFi.

In light of the reviewer's comment, we now have added new experiments performed in different variety of melanoma cell lines reflecting this heterogeneity (Fig. 2h).

Also, it is important to note that these sensitive/resistance signatures have been validated using many cell lines (>20)^{1,10,11} (Fig. 2a-g and Extended data Fig 5), and patient data set (Fig 3)². These two sets of data (cell lines and patients) are now presented in two figures (Fig. 2 and 3), to better evidence that these AhR signatures are a general mechanism.

These *in vitro* and *in vivo* data show a transient induction of the β -signature in response to BRAFi, which is followed by an induction of α - and resistance signatures. These "two waves" (β - followed by α - and resistance signatures) correlate with the dedifferentiation of melanoma cells in response to BRAFi and the selection of resistant cells involved in relapses.

The manuscript has been amended accordingly (L100 – L123) and (L124 – L148) for cell lines and patients respectively.

5. There are no *in vivo* studies showing the utility of targeting the aryl hydrocarbon receptor and that it can prevent the onset of resistance.

We thank the reviewer for this comment improving greatly the manuscript.

As recommended, we performed an *in vivo* experiment to evaluate the protective role of an AhR antagonist here the Resveratrol (RSV). We evaluated the potential of this BRAFi/RSV combination in a patient-derived xenograft (PDX) model of melanoma^{7,12}. Mice were exposed to BRAFi alone or in combination with RSV, once tumors reached 200 mm³. In this model, BRAFi alone stabilized the tumor growth for about 10 days before relapse, while the BRAFi/RSV combination stabilized the tumor for almost 18 days, demonstrating that RSV delays tumor growth significantly (Fig. 6l and Extended data Fig. 8d). In accordance to our *in vitro* results, we demonstrate *in vivo* (PDX model) that targeting AhR with an antagonist is relevant to delay or abrogate relapses in patients.

The manuscript has been amended accordingly (L194 – L199).

6. What is the effect of the BRAF-MEK inhibitor combination on Aryl hydrocarbon receptor signaling? Single agent BRAF inhibition is not routinely used any more and has been superseded by the combination. This would improve the translational impact of these studies.

We agree with the reviewer, standard of care to treat metastatic melanoma patient is now constituted of BRAFi/MEKi double blockade. Thus, to address this critical point, we performed several experiments using MEKi (Cobimetinib and Trametinib) alone or in combination with BRAFi (Vemurafenib and Dabrafenib). We found that MEKi alone and in combination with BRAFi are able to induce genes of the β -signature (Extended data Fig.4b-c).

Moreover, the combination (BRAFi + MEKi) gave similar transcriptional waves (β -signature followed by α - and resistance-signatures) *in vitro* (Extended data Fig.5)⁶, and *in vivo* (Fig. 3g-i, Pt19 and Pt22)² than those observed with BRAFi alone. These results are in

agreement with the literature: almost all patients with metastatic melanoma relapse despite combination (BRAFi + MEKi) but they gained several months when compared to previous treatment (BRAFi alone)^{13,14}.

The manuscript includes now these informations (L135 and L142)

7. Extended Figure 6a (now Extended data Fig. 7a) has no scale. What does the green shading mean?

We apologise for this missing information. To gain in clarity, we have amended the Figure (now Extended data Fig.7a) and removed the gradient of green that was not indispensable for the understanding.

A green square reflects in silico capacity of a defined ligand/molecule to interact with amino acids of the PAS-B domain of AhR. The predicted interactions have been grouped based on their calculated affinity. α and β correspond to the two binding pockets identified on AhR.

These modifications appear now in the figure and figure legend.

8. Minor point: often the English is not good. The paper should be reviewed and corrected by a native English speaker.

To address this particular point, we have used the Nature Research Editing Service. The certificate is attached.

Reviewer #2

BRAF inhibitors are important for treatment of a large subset of cutaneous melanoma patients; however, the response to these agents is limited due to development of drug resistance. This manuscript identified the Ah receptor as a key mediator of BRAFi-resistant genes and shows that resveratrol, a putative Ah receptor antagonist, inhibits BRAFi-resistant genes, confirming a potential role for Ah receptor ligands in combination therapy with BRAF inhibitors for treating cutaneous melanoma. This is an interesting and novel observation; however, the authors need to address the following issues.

1. The overall description of Vem in the first part of Figure 1 could also be interpreted as the action of a novel antagonist of the canonical AhR signaling pathway. The authors should show the combined effects of TCDD+Vem on EROD/luciferase activity, CYP1A1 mRNA levels, DRE binding, and related activities.

We thank the reviewer for this sagacious comment. Indeed, Vem could be considered as a new antagonist of AhR canonical signalling pathway. To further support this claim, we performed competition experiments (see below Supplemental data). We found that exposing cells to Vem (10 μ M) before TCDD (10nM) prevents CYP1A1 mRNA induction. In agreement with these results, Vem abrogates also TCDD-induced AhR transcriptional activity (assessed by 3-XRE-Luc assay (Extended data Fig. 4f). These results are comparable to the ones obtained with the well-known AhR antagonist CH-223191. By electrophoretic mobility shift assay (EMSA), we further demonstrated that Vem competes with TCDD to bind AhR (Extended data Fig. 4g). Together, this tells that Vem acts as an AhR antagonist.

Reciprocally, AhR antagonist such as resveratrol (RSV) abrogates the β -signature induced by Vem (Extended data Fig. 7b).

Altogether, these results strongly suggest that Vem can compete with other AhR ligands to bind AhR. However, Vem is not a classical antagonist since it is also able to induce an AhR-dependent β -signature in an ARNT-independent manner.

Supplemental Data

501Mel cells have been exposed for 2h with Vem (1 μ M) and then treated with TCDD (10 nM) for 24hours. (Positive control for TCDD in Extended data Fig. 4a)

2. The ChIP assay in Extended Figure 2e is somewhat puzzling and requires further explanation. Does this represent constitutive or Vem-induced ligand binding and are similar results observed for both TCDD and Vem?

We thank the reviewer for this comment and apologise for the lack of clarity.

ChIP-PCR experiments were carried out without Vem and TCDD addition. This is now state in the Figure legend. This result represents constitutive AhR/OCA2 regulation. To reinforce the role of AhR on OCA2 expression, we quantified the ability of AhR to transactivate the OCA2 promoter by luciferase assay. In contrast to TCDD, Vem promotes the expression of OCA2 (luciferase activity). These results are in accordance to OCA2 mRNA expression (Extended data Fig. 2) in response to BRAFi.

The GCGTG core binding sites and flanking region traditionally bind AhR:Arnt and AhR binding alone is usually minimal to weak. It would be worth looking at a few AhR coactivators to see if Vem but not TCDD induces their recruitment. Identification of a new partner for the AhR is of prime interest but may be beyond the scope of this manuscript.

In the context of the β -signature in response to Vem, we showed that AhR- gene expression regulation does not require the dimerization with its canonical partner ARNT (TCDD) (Fig. 1c, 1g and Extended data Fig. 2h), which suggests that AhR interacts with non-identified partners. As suggested, the identification of new AhR partners involved in non-canonical AhR binding and activity (β -signature) is of particular interest for the field but beyond the scope of the manuscript.

3. The results of the genomic and functional studies are compelling; however, the authors have ignored the prior studies in this area which show different results. For example,

(i) Leflunomide (an AhR-active pharmaceutical) inhibits melanoma cell growth (PLOS ONE 7, e40926, 2012).

(ii) The AhR has tumor suppressor activity with respect to melanoma growth and metastasis

(iii) Other studies show that the AhR and/or its ligands enhance melanoma growth/invasion

We agree with the reviewer comment and we apologize for this lack. References have been added in the new version of the discussion.

They are referenced as followed:

- (i) : Ref 38 O'Donnell et al 2012 ; Ref 39 Hanson et al., 2018 (L 210)
- (ii) : Ref 36 Roman et al., 2018 ; Ref 37 Narasimhan et al., 2018; Ref 38 O'Donnell et al 2012 ; Ref 39 Hanson et al., 2018 ; Ref 40 Contador-Troca et al., 2015 (L209-210)
- (iii) : Ref 44 Villano et al., 2006 (L214)

4. The authors state the following conclusion: "These results are entirely consistent with recent studies indicating that constitutive and chronic activation of AhR promotes aggressive tumor behavior (31,32) and tumorigenesis in vivo (33, 34)." This statement is incorrect since the AhR and/or its ligand can act as tumor suppressors (colon, breast, pancreatic, ...) or tumor promoters (lung, glioblastoma, ...). The authors should read recent reviews on the AhR and cancer which clearly show these tissue/tumor-specific differences.

We agree with the reviewer comment and we apologize for this shortcut.

The discussion has been toned down and amended with these notions (L211 – L213).

Finally, this study mainly focuses on the long-term effect of AhR's canonical and non-canonical activity in the emergence and acquisition of the resistance mechanism of BRAFi-treated melanomas. We further show that this is associated with dedifferentiated cell state.

1. Tsoi, J., Robert, L., Paraiso, K., Galvan, C., Sheu, K. M., Lay, J., Wong, D. J. L. L., Atefi, M., Shirazi, R., Wang, X., Braas, D., Grasso, C. S., Palaskas, N., Ribas, A. & Graeber, T. G. Multi-stage Differentiation Defines Melanoma Subtypes with Differential Vulnerability to Drug-Induced Iron-Dependent Oxidative Stress. *Cancer Cell* 1–15 (2018). doi:<https://doi.org/10.1016/j.ccell.2018.03.017>
2. Hugo, W., Shi, H., Sun, L., Piva, M., Song, C., Kong, X., Moriceau, G., Hong, A., Dahlman, K. B. B., Johnson, D. B. B., Sosman, J. A. A., Ribas, A. & Lo, R. S. S. Non-genomic and Immune Evolution of Melanoma Acquiring MAPKi Resistance. *Cell* **162**, 1271–1285 (2015).
3. Tirosh, I., Izar, B., Prakadan, S. M., Ii, M. H. W., Treacy, D., Trombetta, J. J., Rotem, A., Rodman, C., Lian, C., Murphy, G., Fallahi-sichani, M., Dutton-regester, K., Lin, J., Kazer, S. W., Gaillard, A. & Kolb, K. E. Dissecting the multicellular ecosystem of metastatic melanoma by single-cell RNA-seq. *Science* (80-.). **352**, 189–196 (2016).
4. Titz, B., Lomova, A., Le, A., Hugo, W., Kong, X., ten Hoeve, J., Friedman, M., Shi, H., Moriceau, G., Song, C., Hong, A., Atefi, M., Li, R., Komisopoulou, E., Ribas, A., Lo, R. S. & Graeber, T. G. JUN dependency in distinct early and late BRAF inhibition adaptation states of melanoma. *Cell Discov.* **2**, 16028 (2016).
5. Shaffer, S. M., Dunagin, M. C., Torborg, S. R., Torre, E. A., Emert, B., Krepler, C., Beqiri, M., Sproesser, K., Brafford, P. A., Xiao, M., Eggan, E., Anastopoulos, I. N., Vargas-Garcia, C. A., Singh, A., Nathanson, K. L., Herlyn, M. & Raj, A. Rare cell variability and drug-induced reprogramming as a mode of cancer drug resistance. *Nature* **546**, 431 (2017).
6. Song, C., Piva, M., Sun, L., Hong, A., Moriceau, G., Kong, X., Zhang, H., Lomeli, S., Qian, J., Yu, C. C., Damoiseaux, R., Kelley, M. C., Dahlman, K. B., Scumpia, P. O., Sosman, J. A., Johnson, D. B., Ribas, A., ... Lo, R. S. Recurrent Tumor Cell-Intrinsic and -Extrinsic Alterations during MAPKi-Induced Melanoma Regression and Early Adaptation. *Cancer Discov.* **7**, 1248 LP-1265 (2017).
7. Rambow, F., Rogiers, A., Marin-Bejar, O., Aibar, S., Femel, J., Dewaele, M., Karras, P., Brawn, D., Hwang Chang, Y., Debiec-Rychter, M., Adriaens, C., Radaelli, E.,

- Wolter, P., Bechter, O., Dummer, R., Levesque, M., Piris, A., ... Marine, J. C. Towards minimal residual disease-directed therapy in melanoma. *Cell in press*, (2018).
8. Sharma, S. V., Lee, D. Y., Li, B., Quinlan, M. P., Takahashi, F., Maheswaran, S., McDermott, U., Azizian, N., Zou, L., Fischbach, M. A., Wong, K.-K., Brandstetter, K., Wittner, B., Ramaswamy, S., Classon, M. & Settleman, J. A chromatin-mediated reversible drug tolerant state in cancer cell subpopulations. *Cell* **141**, 69–80 (2010).
 9. Menon, D. R., Das, S., Krepler, C., Vultur, A., Rinner, B., Schauer, S., Kashofer, K., Wagner, K., Zhang, G., Rad, E. B., Haass, N. K., Soyer, H. P., Gabrielli, B., Somasundaram, R., Hoefler, G., Herlyn, M. & Schaidt, H. A stress-induced early innate response causes multidrug tolerance in melanoma. *Oncogene* **34**, 4545 (2015).
 10. Barretina, J., Caponigro, G. & Stransky, N. The Cancer Cell Line Encyclopedia enables predictive modeling of anticancer drug sensitivity. *Nature* **483**, 603–607 (2012).
 11. Verfaillie, A., Imrichova, H., Atak, Z. K., Dewaele, M., Rambow, F., Hulselmans, G., Christiaens, V., Svetlichnyy, D., Luciani, F., Van den Mooter, L., Claerhout, S., Fiers, M., Journe, F., Ghanem, G.-E., Herrmann, C., Halder, G., Marine, J.-C. & Aerts, S. Decoding the regulatory landscape of melanoma reveals TEADS as regulators of the invasive cell state. *Nat. Commun.* **6**, 6683 (2015).
 12. Gilot, D., Migault, M., Bachelot, L., Journé, F., Rogiers, A., Donnou-Fournet, E., Mogha, A., Mouchet, N., Pinel-Marie, M.-L. M.-L., Mari, B., Montier, T., Corre, S., Gautron, A., Rambow, F., El Hajj, P., Ben Jouira, R., Tartare-Deckert, S., ... Galibert, M.-D. M.-D. A non-coding function of TYRP1 mRNA promotes melanoma growth. *Nat. Cell Biol.* **19**, 1348 (2017).
 13. Ascierto, P. A., McArthur, G. A., Dréno, B., Atkinson, V., Liskay, G., Di Giacomo, A. M., Mandalà, M., Demidov, L., Stroyakovskiy, D., Thomas, L., de la Cruz-Merino, L., Dutriaux, C., Garbe, C., Yan, Y., Wongchenko, M., Chang, I., Hsu, J. J., ... Larkin, J. Cobimetinib combined with vemurafenib in advanced BRAFV600-mutant melanoma (coBRIM): updated efficacy results from a randomised, double-blind, phase 3 trial. *Lancet Oncol.* **17**, 1248–1260 (2016).
 14. Robert, C., Karaszewska, B., Schachter, J., Rutkowski, P., Mackiewicz, A., Stroiakovski, D., Lichinitser, M., Dummer, R., Grange, F., Mortier, L., Chiarion-Sileni, V., Drucis, K., Krajsova, I., Hauschild, A., Lorigan, P., Wolter, P., Long, G. V., ... Schadendorf, D. Improved Overall Survival in Melanoma with Combined Dabrafenib and Trametinib. *N. Engl. J. Med.* **372**, 30–39 (2015).

Reviewers' comments:

Reviewer #1 (Remarks to the Author):

Although the authors have done a generally nice job of addressing most of the initial concerns, one key issue remains that has not been adequately dealt with. One of the key issues raised in the original review was to demonstrate the potential of using RSV as a strategy to target AhR and prevent BRAF inhibitor resistance in vivo. Although these studies have now been included they are a little unconvincing. Although there is a trend demonstrating that RSV may enhance the effects of BRAF inhibition, the results are quite weak with huge error bars. The data is presented in a non-traditional way, where the tumor growth curves are representing the best fit trendline of the remaining live mice only. It is more customary, in solid tumor studies, to show the average tumor growth curves (not trendlines) for all animals at all time points. When animals start reaching endpoints, the experiment is terminated. Following this logic, the data is really only adequate up to day ~14 where all mice are still in the experiment. With this in mind, the average difference in tumor volume at this time point does not appear biologically significant (~400 mm³ vs 550 mm³). These kinds of tumor growth curves are usually also drawn to represent the average tumor size at each time point, not the general "best fit" style trendline shown in the manuscript. It is not scientifically appropriate to continue a "trend" when most of the mice have already dropped out, of course the tumor volumes would improve when all of the tumor growth from mice with large tumors are excluded, presumably because they have reached endpoint. I am not sure that these data significantly support the idea that targeting AhR will be useful as a strategy to restrain BRAF inhibitor resistance.

Reviewer #2 (Remarks to the Author):

The authors have addressed most of my critique with the exception of their brief revised discussion of the role of the AhR in carcinogenesis (#3, Reviewer 2). Most studies clearly demonstrate that the AhR is eminently druggable for cancer chemotherapy however the endogenous role of the AhR is not "still a matter of debate" (last paragraph) for some cancers as indicated previously. This should be further clarified.

NCOMMS-18-02506-A
Answers to Reviewers

We first would like to thank the reviewers for reviewing the revised version of the manuscript entitled "Sustained activation of the Aryl hydrocarbon Receptor transcription factor promotes the resistance to BRAF inhibitors in melanoma".

We are happy to see that the reviewers appreciate the work provided in the revised version.

Please see below how we addressed the remaining 2 pending points.

Reviewer #1 :

Although the authors have done a generally nice job of addressing most of the initial concerns, one key issue remains that has not been adequately dealt with. One of the key issues raised in the original review was to demonstrate the potential of using RSV as a strategy to target AhR and prevent BRAF inhibitor resistance in vivo. Although these studies have now been included they are a little unconvincing. Although there is a trend demonstrating that RSV may enhance the effects of BRAF inhibition, the results are quite weak with huge error bars. The data is presented in a non-traditional way, where the tumor growth curves are representing the best fit trendline of the remaining live mice only. It is more customary, in solid tumor studies, to show the average tumor growth curves (not trendlines) for all animals at all time points. When animals start reaching endpoints, the experiment is terminated. Following this logic, the data is really only adequate up to **day ~14** where all mice are still in the experiment. With this in mind, the average difference in tumor volume at this time point does not appear biologically significant (~400 mm³ vs 550 mm³). These kinds of tumor growth curves are usually also drawn to represent the average tumor size at each time point, not the general "best fit" style trendline shown in the manuscript. It is not scientifically appropriate to continue a "trend" when most of the mice have already dropped out, of course the tumor volumes would improve when all of the tumor growth from mice with large tumors are excluded, presumably because they have reached endpoint. I am not sure that these data significantly support the idea that targeting AhR will be useful as a strategy to restrain BRAF inhibitor resistance.

We thank the reviewer for this final remark that will clarify this section of the manuscript. To address this point we have formatted in a classical way, as recommended, the *in vivo* data (Fig. 6l-6m), by generating two graphs:

- Fig. 6l, recapitulates "Tumor Volume" in mm³ at **day 14**, as requested, according to the treatment used (BRAFi alone or BRAFi/RSV).
- Fig. 6m, recapitulates the **Number of Days to reach** a tumor volume > 800mm³, bearing in mind that 1000mm³ corresponds to the **ethical end-point**.
- The original Fig. 6l (previous version) is now presented as a supplementary data (Supplementary Fig. 8c) and has been modified taking in account the reviewer remark. The time course is limited in days (18 days) maintaining a minimal number of mice (n=5) per arm (BRAFi alone / BRAFi + RSV) and we now show the **average tumor growth**, as recommended.

The main text has been modified:

Original version, June 2018 (L197-202), the green section has been modified in the new version: " We evaluated the potential of this BRAFi/RSV combination in a patient-derived xenograft (PDX) model of melanoma ^{23,31}. Mice were exposed to BRAFi alone or in

combination with RSV once tumors reached 200 mm³. In this model, BRAFi alone stabilized the tumor growth for about 10 days before relapse, while the BRAFi/RSV combination stabilized the tumor for almost 18 days, demonstrating that RSV delays tumor growth significantly (Fig. 6l and Extended data Fig. 8d). “

Revised version July 20, 2018 (L195-203): “We evaluated the potential of this BRAFi/RSV combination in a patient-derived xenograft (PDX) model of melanoma^{23,31}. Mice were exposed to BRAFi alone or in combination with RSV once tumors reached 200 mm³. In this model, BRAFi alone stabilized the tumor growth for about 10 days before relapse (Extended data Fig. 8d). BRAFi/RSV treatment reduced, moderately but significantly, tumor growth compared to BRAFi alone (14 days) (Fig. 6l). In agreement with these results, tumors exposed to BRAFi/RSV combination reached endpoints significantly later than BRAFi alone (24 and 16 days, respectively) (Fig. 6m). Together these results support the use of AhR inhibitors to increase BRAFi efficacy over-time. “

The conclusion has also been modified (L219, removing “such as RSV”), to open-up on the use of AhR-antagonist and not on RSV only, bearing in mind that AhR-antagonists developed by Hercules Pharmaceutical are in pre-clinical development.

Reviewer #2 (Remarks to the Author):

The authors have addressed most of my critique with the exception of their brief revised discussion of the role of the AhR in carcinogenesis (#3, Reviewer 2). Most studies clearly demonstrate that the AhR is eminently druggable for cancer chemotherapy however the endogenous role of the AhR is not "still a matter of debate" (last paragraph) for some cancers as indicated previously. This should be further clarified.

We thank the reviewer for this final remark.

To clarify this point we have modified the text as followed:

“Indeed, AhR is now reported to have pro- or anti- tumor activity according to cell state” (L213).

To support this point the following papers are quoted:

34. Roman, A. C., Carvajal-Gonzalez, J. M., Merino, J. M., Mulero-Navarro, S. & Fernández-Salguero, P. M. The aryl hydrocarbon receptor in the crossroad of signalling networks with therapeutic value. *Pharmacol. Ther.* **185**, 50–63 (2018).
35. Narasimhan, S., Stanford, E., Novikov, O., Parks, A., Schlezinger, J., Wang, Z., Laroche, F., Feng, H., Mulas, F., Monti, S. & Sherr, D. Towards Resolving the Pro- and Anti-Tumor Effects of the Aryl Hydrocarbon Receptor. *Int. J. Mol. Sci.* Accepted (2018). doi:10.3390/ijms19051388
36. O’Donnell, E. F., Kopparapu, P. R., Koch, D. C., Jang, H. S., Phillips, J. L., Tanguay, R. L., Kerkvliet, N. I. & Kolluri, S. K. The Aryl hydrocarbon receptor mediates leflunomide-induced growth inhibition of melanoma cells. *PLoS One* **7**, (2012).
37. Hanson, K., Robinson, S. R., Al-Yousuf, K., Hendry, A. E., Sexton, D. W., Sherwood,

- V. & Wheeler, G. N. The anti-rheumatic drug, leflunomide, synergizes with MEK inhibition to suppress melanoma growth. *Oncotarget* **9**, 3815–3829 (2018).
38. Contador-Troca, M., Alvarez-Barrientos, A., Merino, J. M., Morales-Hernández, A., Rodríguez, M. I., Rey-Barroso, J., Barrasa, E., Cerezo-Guisado, M. I., Catalina-Fernández, I., Sáenz-Santamaría, J., Oliver, F. J. & Fernandez-Salguero, P. M. Dioxin receptor regulates aldehyde dehydrogenase to block melanoma tumorigenesis and metastasis. *Mol. Cancer* **14**, 148 (2015).